# RAGCHECKER: A Fine-grained Framework for Diagnosing Retrieval-Augmented Generation

**Dongyu Ru**[1*]   **Lin Qiu**[1*]   **Xiangkun Hu**[1*]   **Tianhang Zhang**[1*]   **Peng Shi**[1*]
**Shuaichen Chang**[1*]   **Cheng Jiayang**[1†]   **Cunxiang Wang**[1†]   **Shichao Sun**[1]
**Huanyu Li**[2]   **Zizhao Zhang**[1†]   **Binjie Wang**[1†]   **Jiarong Jiang**[1]   **Tong He**[1]
**Zhiguo Wang**[1]   **Pengfei Liu**[2]   **Yue Zhang**[3]   **Zheng Zhang**[1]

[1]Amazon AWS AI   [2]Shanghai Jiaotong University   [3]Westlake University

## Abstract

Despite Retrieval-Augmented Generation (RAG) showing promising capability in leveraging external knowledge, a comprehensive evaluation of RAG systems is still challenging due to the modular nature of RAG, evaluation of long-form responses and reliability of measurements. In this paper, we propose a fine-grained evaluation framework, RAGCHECKER, that incorporates a suite of diagnostic metrics for both the retrieval and generation modules. Meta evaluation verifies that RAGCHECKER has significantly better correlations with human judgments than other evaluation metrics. Using RAGCHECKER, we evaluate 8 RAG systems and conduct an in-depth analysis of their performance, revealing insightful patterns and trade-offs in the design choices of RAG architectures. The metrics of RAGCHECKER can guide researchers and practitioners in developing more effective RAG systems[3].

## 1   Introduction

Retrieval-Augmented Generation (RAG) systems [18, 7] enhance Large Language Models (LLMs) by incorporating external knowledge bases, enabling more precise and contextually relevant responses [7, 53, 13]. As these systems become integral to a variety of applications [54, 2, 8], it's imperative to develop robust and comprehensive evaluation frameworks to assess their performance and identify areas for improvement. Evaluating RAG systems, however, presents several challenges:

(1) *modular complexity*: The modular nature of RAG systems, comprising both a retriever and a generator, complicates the design of effective evaluation metrics. It is crucial to establish metrics that can holistically assess the entire system as well as evaluate the individual modules and their interplay [53], allowing for fully understanding the sources of the errors and misses and how they are generated. (2) *metric limitation*: Existing metrics for evaluating RAG systems, which are often rule-based or coarse-grained, fall short in providing accurate and interpretable results. Specifically, traditional metrics like recall@k and MRR [44] for retrievers depend on annotated chunks and a rigid chunking approach, missing out on the full semantic scope of the knowledge base. For generators, typical measures such as n-gram-based (e.g., BLEU [30], ROUGE [19]), embedding-based (e.g., BERTScore [56]), and LLM-based methods [45] perform well with concise answers but fail to detect finer distinctions in longer responses. To bridge these gaps, it is essential to develop detailed, semantic-based evaluation metrics that effectively capture the intricacies and overall quality of both the retrieval and generation components in RAG systems. (3) *metric reliability*: the reliability

---

[*]Shared first authorship.

[†]Work done during internship at Amazon.

[3]This work has been open sourced at `https://github.com/amazon-science/RAGChecker`

of existing metrics for RAG remains under-explored. Effective evaluation metrics must not only accurately reflect system performance but also align with human judgments to ensure their utility in real-world scenarios.

To overcome these challenges, we introduce RAGCHECKER, an innovative evaluation framework designed for detailed analysis of both retrieval and generation processes. RAGCHECKER is based on claim-level entailment checking which involves operations of extracting claims from the response and ground truth answer and checking them against other texts. This approach enables fine-grained evaluation instead of response-level assessment. RAGCHECKER processes the user query, retrieved context, response, and ground truth answer, producing a suite of metrics:

1. **Overall Metrics** to provide a holistic view of the system performance, assessing the overall quality of the generated responses.
2. **Diagnostic Retriever Metrics** to evaluate the effectiveness of the retriever, identifying its strengths and weaknesses in finding relevant information from the knowledge base.
3. **Diagnostic Generator Metrics** to assess the performance of the generator, diagnosing how well the generator utilizes the retrieved context, handles noisy information, and generates accurate and faithful responses.

Compared to existing evaluation frameworks, RAGCHECKER provides a more comprehensive assessment of RAG systems. While some frameworks offer fine-grained evaluation only on certain metrics (e.g., RAGAS [5], TruLens [6], ARES [35]) or evaluate specific aspects of RAG (e.g., RGB [4], RECALL [22], NoMIRACL [40]), RAGCHECKER's metrics are all based on fine-grained claim-level checking and are designed to provide actionable insights into the sources of errors.

To ensure the reliability of RAGCHECKER, we annotate a human judgment dataset to assess the correlations between the proposed metrics and human judgments. This meta-evaluation validates the effectiveness of RAGCHECKER in capturing the quality and reliability of RAG systems from a human perspective. We demonstrate the effectiveness of RAGCHECKER through comprehensive experiments evaluating 8 state-of-the-art RAG systems on a benchmark repurposed from public datasets across 10 domains. In-depth analysis of the evaluation results reveals that RAGCHECKER provides insightful diagnostic signals (Sec. 4.3) pointing the directions for improvements of RAG systems (Sec. 4.4).

The main contributions of this paper are as follows:

- We propose RAGCHECKER, a novel RAG evaluation framework that offers fine-grained evaluation for both the retriever and generator components, introducing new diagnostic metrics to provide actionable insights into the sources of errors.
- We conduct meta evaluation and verified RAGCHECKER has significantly better correlations with human judgements than other evaluation metrics.
- We perform extensive experiments evaluating 8 RAG systems on our curated benchmark across 10 domains, and uncover valuable insights, such as the trade-off between retrieval improvement and noise introduction, and the tendency of faithful open-source models to blind trust on context.

## 2 Related Work

### 2.1 Retrieval Augmented Generation

Large Language Models (LLMs) demonstrate strong capabilities in generating text, but there are also obstacles such as outdated information and the potential to hallucinate [42, 46, 12]. To address these issues, RAG retrieves external knowledge to generate responses with improved accuracy and factuality [7, 53, 13]. Integrating external knowledge is especially crucial in fields like legal, medical and finance, where precision and reliability are essential [24, 50, 55].

RAG systems have shown impressive performance across a range of tasks, including open-domain question answering [27, 10, 18], code generation [32, 57, 38] and dialogue [37, 16, 41]. Additionally, real world products like Bing Search[4] and Langchain [3] have integrated applications based on RAG.

---

[4]`https://www.bing.com/chat`

## 2.2 Evaluation of RAG

Existing evaluation practices for RAG systems can be categorized into two main approaches: evaluating essential capabilities of generators only and assessing end-to-end performance of RAG systems.

Within the two components of a RAG system, the retriever has been well studied in recent years, thus a line of recent work focused on evaluating essential generator capabilities. RGB [4] evaluated 4 fundamental abilities required for generators including Noise Robustness, Negative Rejection, Information Integration and Counterfactual Robustness by manually constructed test sets. RECALL [22] introduced manually edited counterfactual contexts into QA and text generation datasets to evaluate the counterfactual robustness of LLMs. NoMIRACL [40] evaluated LLMs' robustness against first-stage retrieval errors of RAG systems with manually judged relevant and non-relevant datasets. Wu et al. [49] quantified the tug-of-war between LLMs' faithfulness and internal prior by introducing varying levels of perturbations on the provided contexts. FaaF [15] introduced a fine-grained fact verification formulation to improve previous prompting-based approaches in evaluating factuality of generators. However, we argue that above generator-only evaluation approaches with manually constructed datasets cannot serve as a general RAG evaluation framework to reveal the entanglement of between generation results and different retrieval behaviors, as shown in the analysis of Sec. 4.3.

Another line of work focused on assessing end-to-end quality scores of RAG systems. TruLens [6] introduced the concept of RAG Triad, which decompose the quality scores into three aspects: context relevance, groundedness and answer relevance, then predicted the score by prompting LLMs or using NLI models. RAGAS [5] and ARES [35] followed the RAG Triad concept and improved the score prediction approaches on different datasets. CRUD-RAG [25] refered to the CRUD (Create, Read, Update and Delete) actions between users and knowledge bases to develop corresponding datasets and evaluation metrics for RAG systems. We compare the above four evaluation frameworks with RAGCHECKER in the meta evaluation of Sec. 4.2.

Besides, the following work also provided good insight or high quality datasets for end-to-end RAG evaluation. Liu et al. [21] conducted human evaluation to audit four popular generative search engines in terms of fluency, perceived utility, and verifiability. MEDRAG [50] constructed a medical RAG benchmark from medical QA datasets and evaluated medical RAG systems with QA accuracy. MultiHop-RAG [39] generated multi-hop queries from news articles and evaluated RAG systems with QA accuracy. CDQA [52] proposed a novel approach to generate dynamic QA questions which requires latest information to answer. However, the evaluation metrics used in the work mentioned above rely either on human evaluation or simple textual accuracy, making them incapable of complex RAG scenarios that require long answer evaluation. Therefore, we do not include them in the meta evaluation.

## 3   RAGCHECKER Framework

**Formulation**   Define a modular RAG system as RAG = {R, G}, where R is the retriever and G is the generator. Given a query $q$ and documents $D$, it first retrieves top-$k$ relevant context $\{\text{chunk}_j\} = \text{R}(q, D, k)$, and then generates a model response m = G($\{\text{chunk}_j\}, q$). For simplicity, we can also represent the overall RAG generation process as m = RAG($q, D$).

**Design Principle**   Given the compositional nature of RAG, we observe there are two major personae using a RAG evaluation framework. The first persona is a user that cares about the overall performance of RAGs and might choose a system with the best performance. Such a persona prefers a single value metric to compare and rank among RAG systems against a benchmark. The second persona is a developer that focuses on improving a RAG system with the need to identify causes of mistakes and potential rooms for improvements. Causes of errors in response can be classified into 1) retrieval errors, where the retriever fails to return complete and relevant context, and 2) generator errors, where the generator struggles to identify and leverage relevant information from context.

Consequently, metrics that reveal error causes should be different from those for overall performance, in the sense that error causes are module-specific or even reflected only by a certain behavior of a module. To help both personae to assess RAG performance, we design RAGCHECKER , a evaluation framework of RAG systems that consists of a benchmark with rich annotations and a set of diversely-purposed fine-grained metrics.

### 3.1 Inputs to RAGCHECKER

We prepare each sample in our benchmark dataset in the format of a tuple $\langle q, D, gt \rangle$ representing query, documents, and ground-truth answer, where query is the input question to a RAG system, documents form the database providing possible context and are processed into chunks with the same number of tokens, and ground-truth answer is a complete and correct answer for the input question. Further information is provided in Sec. 4.1.

### 3.2 Fine-grained Evaluation with Claim Entailment

As illustrated in Fig. 1, a response generated by a RAG system might be a mixture of correct ( ○ ) and incorrect claims ( ✗ ), while also missing some in-ground-truth claims ( △ ). In this sense, evaluating responses at a finer granularity is crucial to comprehensively assess the quality of an answer. For this purpose, we introduce two components: 1) a text-to-claim extractor that decomposes a given text $T$ into a set of claims $\{c_i\}$, and 2) a claim-entailment checker to determine whether a given claim $c$ is entailed ($\in$) in a reference text $Ref$ or not ($\notin$).

### 3.3 RAGCHECKER Metrics

With the annotation and claim-level entailment functions specified, we next define the metrics. For a RAG user, we design metrics to compare the performance among RAG systems, including a single-value F1 score as an overall metric. For a RAG developer, on the other hand, we propose two sets of modular metrics for the retriever and the generator in a RAG system respectively, that aim to decompose the system and diagnose the source of errors. In the rest of this section, we will first introduce the overall metrics and then go over modular metrics for retriever and generator separately. The formulas for each metric are summarized in Appendix B.

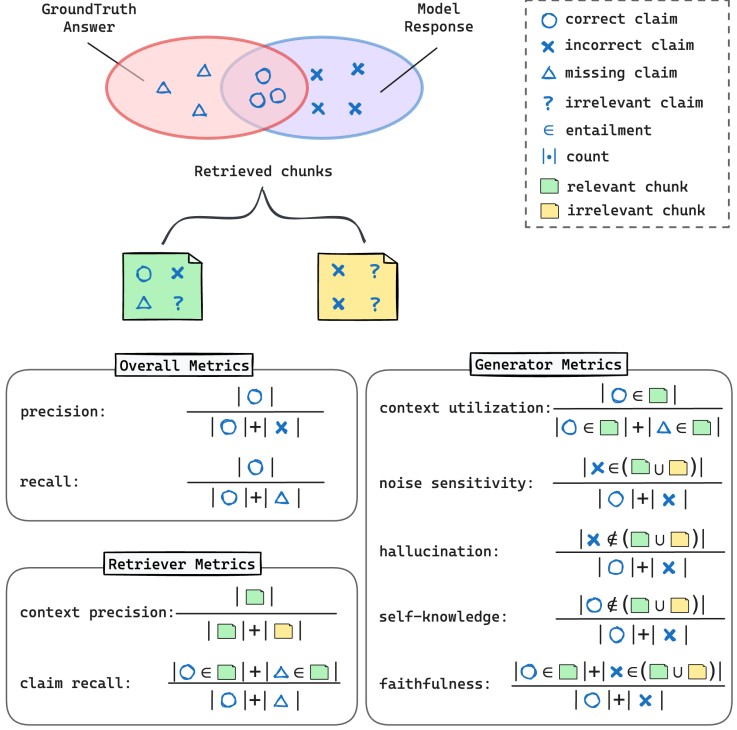

Figure 1: Illustration of the proposed metrics in RAGCHECKER . The upper Venn diagram depicts the comparison between a model response and the ground truth answer, showing possible correct( ○ ), incorrect( ✗ ), and missing claims( △ ). The retrieved chunks are classified into two categories based on the type of claims they contain. Below, we define the overall, retriever, and generator metrics, illustrating how each component of the RAG system is evaluated for its performance.

### 3.3.1 Overall Metrics

To assess the overall response quality of a RAG system from a user's perspective, we can compute the precision and recall at claim level for each model generated response against its paired ground-truth answer. Specifically, we first extract claims from a model response $m$ and a ground-truth answer $gt$ as $\{c_i^{(m)}\}$ and $\{c_i^{(gt)}\}$ respectively. Then, we define correct claims in the response as $\{c_i^{(m)}|c_i^{(m)} \in gt\}$, and correct claims in the ground-truth answer as $\{c_i^{(gt)}|c_i^{(gt)} \in m\}$. Two metrics can be computed directly: **precision** is the proportion of correct claims in all response claims, and **recall** is the proportion of correct claims in all ground-truth answer claims. Further, the harmonic average of precision and recall gives the **F1** score, as the overall performance metric.

### 3.3.2 Retriever Metrics

Ideally, a perfect retriever returns precisely all claims needed to generate the ground-truth answer. Completeness-wise, we can measure how many claims made in the ground-truth answer are covered by retrieved chunks. With retrieved chunks as the reference text, we compute **claim recall** as the proportion of $\{c_i^{(gt)}|c_i^{(gt)} \in \{\text{chunk}_j\}\}$.

Differently, we define the retriever precision at chunk-level instead of claim-level. A retrieved chunk is called *relevant chunk* (r-chunk), if any ground-truth claim is entailed in it. In other words, $\text{chunk}_j$ is a relevant chunk if $\exists i, \ s.t. \ c_i^{(gt)} \in \text{chunk}_j$. The rest retrieved chunks are called *irrelevant chunk* (irr-chunk). The retriever's **context precision** is defined as $|\{\text{r-chunk}_j\}|/k$, where $k$ is the number of all retrieved chunks.

Note that a chunk-level precision provides better interpretability than a claim-level one, because in practice RAG systems usually work with documents processed to be text chunks in a fixed size. That being said, it is likely that a chunk may contain relevant claims and irrelevant or misleading information at the same time. As a result, the best possible retriever can only achieve a claim-level precision score lower than 100%, and such an upper-bound varies depending on the actual text distribution in $D$ and chunking strategy.

### 3.3.3 Generator Metrics

Given $k$ retrieved chunks (possibly mixing relevant and irrelevant information), a perfect generator would identify and include all ground-truth-relevant claims and ignore any that are not. Because the generator's results have dependency on retrieved chunks, we provide in total six metrics characterizing different aspects of its performance.

Given a model response $m$ and its claims $\{c_i^{(m)}\}$, we first compute the proportion of $c_i^{(m)}$ that are entailed in retrieved chunks. This metric is **faithfulness**, as it describes how faithful the generator is to the provided context, thus the higher the better.

Next, we examine three types of incorrect response claims, i.e. $\{c_i^{(m)}|c_i^{(m)} \notin gt\}$.

1. The first type includes incorrect claim that are entailed in a relevant chunk, then it indicates the generator is sensitive to noise coupled with useful information. The proportion of this type of claims to all $\{c_i^{(m)}\}$ is **relevant noise sensitivity**.
2. The second type includes incorrect claim that are entailed in an irrelevant chunk, then it indicates the generator is also sensitive to noise even in an irrelevant context. The proportion of these incorrect claims is **irrelevant noise sensitivity**.
3. Finally, the third type includes incorrect claims that are not entailed in any retrieved chunk, meaning all such claims are generated by the generator itself. Its proportion is **hallucination**.

Note that for simplicity we group the two noise sensitivities in Fig. 1, but later in Sec. 4.3 we can see that generators generally has different sensitivity to relevant and irrelevant noise.

Finally, we characterize how a generator uses information sources to produce correct claims. A correct claim not entailed by any chunk can only be based on generator's **self-knowledge**, thus the proportion of these claims reflects how many correct claims are generated on its own. A lower self-knowledge score is better, when the generator is expected to fully depend on retrieved context only in a RAG system. On the other hand, we also check how much retrieved relevant information is

used by the generator. Retrieved relevant information is measured by the number of ground-truth answer claims entailed in retrieved chunks, while the evidence of being used by generator is reflected by entailment in model response. Therefore, the **context utilization** is computed as the ratio between $|\{c_i^{(gt)}|c_i^{(gt)} \in \{\text{chunk}_j\} \text{ and } c_i^{(gt)} \in m\}|$ and $|\{c_i^{(gt)}|c_i^{(gt)} \in \{\text{chunk}_j\}\}|$. Generally a higher context utilization is preferred.

## 4 Experiments

### 4.1 Experimental Setup

**Baseline RAG Systems** We apply RAGCHECKER to 8 customized RAG systems to demonstrate how these metrics reflect the properties and differences among them, and how they guide the refinement of these systems. The 8 RAG systems are combinations with 2 retrievers and 4 generators. For retrievers, we choose BM25 [33], a representative classic sparse retrieval framework, and E5-Mistral [48], the SOTA open-source dense retriever. Our four generators are GPT-4 [29], Mixtral-8x7B [14], Llama3-8B, and Llama3-70B [1], covering open-source and proprietary LLMs in various sizes. Further details are deferred to Appendix D. We employ Llama3-70B as both the claim extractor and checker models implemented by an open-sourced framework RefChecker[5] [11]. As a validation of its performance on the RefChecker's hallucination detection benchmark, this setup outperforms the best purely open-sourced combinations reported in RefChecker's paper (see Appendix G).

**Benchmark Datasets** For comprehensive evaluations, we curate a benchmark containing 4,162 queries across 10 domains. This benchmark is repurposed from public datasets of open domain question answering, spanning domains of Wikipedia, AI science, novel, biomedical, finance, lifestyle, recreation, science, technology and writing. We convert the short answers to long-form answers in the datasets to align with the current LLM-based RAG systems. Please refer to Appendix A for the details of the benchmark curation process. The statistics of the benchmark are shown in Tab. 1.

### 4.2 Meta Evaluation

We first conduct the meta evaluation to verify the soundness of RAGCHECKER and compare with existing baseline RAG evaluation frameworks.

**Baseline RAG Evaluation Frameworks** We include a total of 10 metrics from Trulens [6], RA-GAS [5], ARES [35] and CRUD-RAG [25] in the meta evaluation, as they are capable to evaluate end-to-end performance with long answers. Metrics selected for comparison along with their descriptions are summarized in Tab. 4 of Appendix C. To ensure a fair comparison, we use Llama3-70B-Instruct as the LLM backbone when applicable. Since models in the Llama3 family don't provide an embedding model, baseline metrics requiring embedding capability still use their corresponding default LLM backbones. In addition to the 10 metrics detailed in the table, we also incorporate BLEU [31], ROUGE-L [20], and BERTScore [56] to assess the correlation between the generated responses and the ground truth answers.

**Meta Evaluation Dataset** All baseline metrics are designed with different aspects and functionalities to a certain degree, thus making an exact comparison over metric scores inapplicable. However, we argue that a good metric should reflect the relative human preference over different RAG systems. In this spirit, we construct the meta evaluation dataset with sampled instances from the generated responses of 8 baseline RAG systems introduced in Sec. 4.1 on our benchmark. Each meta evaluation instance is a pair of responses from two baseline RAG systems given the same query. By considering all combinations over 10 domains and 28 baseline pairs, we end up with 280 instances for pairwise human preference labeling. For each instance, annotators compare a pair of responses based on correctness, completeness, and overall assessment. For each aspect, annotators measure their preferences as one out of five relative choices, including significantly better (2), slightly better (1), tie (0), slightly worse (-1) and significantly worse (-2). For quality control, each instance is annotated by two annotators, and their overall agreement and correlation are measured. To conclude, we build a meta evaluation dataset with 280 instances, each instance is labeled by two annotators with their preference in terms of correctness, completeness and overall assessment.

---

[5]https://github.com/amazon-science/RefChecker

Table 1: Statistics of the RAG benchmark. This benchmark is repurposed from public datasets across 10 domains, containing 4,162 questions. For the domains of Finance, Lifestyle, Recreation, Technology, Science and Novel, the short answers are extended to long-form answers with GPT-4.

| Dataset | Domain | # Query | # Doc. | Source | Example Query |
|---|---|---|---|---|---|
| ClapNQ | Wikipedia | 300 | 4,293 | ClapNQ | Difference between russian blue and british blue cat |
| NovelQA | Novel | 280 | 19 | NovelQA | When do the Ewell kids go to school? |
| RobustQA Writing | Writing | 500 | 199,994 | LoTTE, RobustQA | What is the difference between online and internet? |
| RobustQA BioASQ | Biomedical | 511 | 197,816 | BioASQ | What hand deformities do patients with Apert syndrome present with? |
| RobustQA Finance | Finance | 500 | 57,638 | FiQA, RobustQA | Is it safer to send credit card number via unsecured website form or by e-mail? What safer options are there? |
| RobustQA Lifestyle | Lifestyle | 500 | 119,461 | LoTTE, RobustQA | Can i eat a day old peanut butter sandwich? |
| RobustQA Recreation | Recreation | 500 | 166,975 | LoTTE, RobustQA | Why are so many american (spy) movies set in europe? |
| RobustQA Science | Science | 500 | 125,368 | LoTTE, RobustQA | Where is the flaw in this proof that 1=2? (derivative of repeated addition) |
| RobustQA Technology | Technology | 500 | 638,509 | LoTTE, RobustQA | Why not use larger cipher keys? |
| KIWI | AI Science | 71 | 429 | KIWI | What are the prior approaches proposed to improve faithfulness of the reasoning steps generated by LLMs and what tasks are they applied on? |

**Meta Evaluation Process and Results** Based on the meta evaluation dataset, we perform the following evaluation process. Since the human preference labels can be seen as the score difference of a response pair: $h_i \in \{-2, -1, 0, 1, 2\}$, with a baseline RAG evaluation model $E$, we compute a normalized score difference as $e_i = f(E(r_i^2) - E(r_i^1)) \in [-2, 2]$, where $f$ is a linear normalization function. Our meta evaluation is the correlation between $h_i$ and $e_i$ overall 280 instances as reported in Tab. 2, together with the correlation between $h_i$ and $h'_i$ from two annotators as the upper-bound. In addition, we further compute human agreement rate as the proportion of instances satisfying $abs(h_i - h'_i) \le 1$, and the result is 90.95%.

Table 2: Correlation results with Human Evaluation of Correctness, Completeness, and Overall Assessment. We only show the metric with the best correlation for each baseline framework. Full results can be found in Tab. 5 of Appendix C.

| Baseline | Metric | Correctness | | Completeness | | Overall Assessment | |
|---|---|---|---|---|---|---|---|
| | | Pearson | Spearman | Pearson | Spearman | Pearson | Spearman |
| BLEU | BLEU-avg | 38.89 | 35.32 | 32.13 | 21.85 | 35.14 | 29.42 |
| ROUGE | ROUGE-L | 31.75 | 31.72 | 47.88 | 45.67 | 43.10 | 43.21 |
| BERTScore | BERTScore | 30.34 | 27.05 | 37.93 | 40.05 | 33.51 | 35.57 |
| TruLens | Answer Relevance | 35.01 | 27.37 | 37.24 | 37.91 | 35.15 | 33.59 |
| ARES | Answer Relevance | 18.63 | 16.84 | 20.13 | 18.13 | 17.81 | 16.26 |
| RAGAS | Answer Similarity | 41.07 | 43.21 | 53.16 | **61.35** | 48.31 | 57.23 |
| CRUD-RAG | Recall | 30.93 | 27.13 | 45.11 | 43.76 | 41.25 | 39.71 |
| RAGChecker | Same metric as human | **49.66** | **46.95** | **60.67** | 58.11 | **61.93** | **60.90** |
| Human | Annotator correlation | 63.67 | 59.19 | 71.91 | 68.36 | 70.09 | 68.89 |

From the table, we can observe that RAGCHECKER has the strongest correlation with human preference in terms of three aspects. Among other baseline metrics, Answer Similarity of RAGAS, which is based on the stronger backbone model text-embedding-ada-002 [28], shows the best performance. We also provide a detailed comparison between RAGCHECKER and this strongest baseline in Fig. 4

of Appendix C. As an upper bound, the human correlations at the bottom show that there is still a clear gap between model predictions and human annotators.

## 4.3 Main Results

We present the averaged evaluation results for 8 RAG systems across 10 diverse domain datasets in Tab. 3. Additional results for all datasets are provided in Appendix E. The RAG system that exhibited the best performance in our experiments is E5-Mistral_GPT-4, owing to the strong retrieval capability of E5-Mistral coupled with the adept comprehension abilities of GPT-4. Next, we provide a list of insights induced from Tab. 3, along with their interpretation and possible directions for improvements.

Table 3: The averaged evaluation results for different RAG systems across 10 datasets. The overall performance of the RAG system is quantified using precision (Prec.), recall (Rec.), and F1 scores. The retriever component is evaluated based on claim recall (CR) and context precision (CP), while the generator component is diagnosed through context utilization (CU), relevant noise sensitivity (NS(I)), irrelevant noise sensitivity (NS(II)), hallucination (Hallu.), self-knowledge (SK), and faithfulness (Faith.). Additionally, the average number of response claims for each RAG system is provided.

| RAG systems | Overall | | | Retriever | | Generator | | | | | | #Claim |
|---|---|---|---|---|---|---|---|---|---|---|---|---|
| | Prec.↑ | Rec.↑ | F1↑ | CR↑ | CP↑ | CU↑ | NS(I)↓ | NS(II)↓ | Hallu.↓ | SK↓ | Faith.↑ | |
| BM25_GPT-4 | 61.0 | 49.7 | 50.3 | 74.0 | 52.3 | 61.4 | 26.2 | 4.1 | 8.7 | 3.4 | 87.9 | 12 |
| BM25_Llama3-8b | 52.1 | 43.9 | 42.1 | 74.0 | 52.3 | 54.9 | 31.3 | 6.1 | 9.8 | 1.8 | 88.4 | 11 |
| BM25_Llama3-70b | 59.1 | 44.9 | 46.3 | 74.0 | 52.3 | 56.2 | 30.4 | 5.3 | 5.1 | 1.7 | 93.2 | 9 |
| BM25_Mixtral-8x7b | 52.5 | 44.3 | 42.9 | 74.0 | 52.3 | 54.9 | 34.3 | 5.8 | 6.2 | 1.8 | 92.0 | 9 |
| E5-Mistral_GPT-4 | 62.0 | 53.0 | 52.7 | 83.5 | 61.8 | 60.4 | 28.9 | 3.5 | 5.7 | 1.4 | 92.9 | 12 |
| E5-Mistral_Llama3-8b | 53.8 | 48.3 | 45.0 | 83.5 | 61.8 | 55.0 | 33.5 | 5.5 | 6.6 | 0.8 | 92.7 | 11 |
| E5-Mistral_Llama3-70b | 60.6 | 50.4 | 50.2 | 83.5 | 61.8 | 57.6 | 31.7 | 4.3 | 3.3 | 0.8 | 95.9 | 10 |
| E5-Mistral_Mixtral-8x7b | 53.1 | 48.6 | 45.7 | 83.5 | 61.8 | 55.2 | 36.5 | 5.1 | 4.0 | 0.8 | 95.2 | 10 |

**Retriever Matters Consistently.** The quality of retrieval is crucial, as evidenced by the notable differences in overall Precision, Recall and F1 scores when comparing BM25 with E5-Mistral with the generator fixed. This improvement is agnostic to the specific choice of generator, suggesting a consistent benefit from employing a better retriever.

**Generator Model Size Brings All-Round Improvement.** Paired to the same retriever, Llama3-70B consistently achieves better overall performance than Llama3-8B. More concretely, this superiority is supported by a better performance over every generator metric, such as improved context utilization, reduced noise sensitivity, and less hallucination.

**Stable and Performant Context Utilization is Key.** Among all generator metrics, we observe that context utilization strongly correlates to the overall F1 score, while such correlation is relatively weaker for other generator metrics. Also, generators' context utilization are relatively stable between the two retrievers, meaning their overall recall can be improved with a better retriever. These observations indicate that the capability to fully utilize retrieved context is key, which is intuitive because the generator in a RAG system is expected to leverage context to surpass its self-knowledge.

**Informative Context Improves Faithfulness and Reduces Hallucination.** As E5-Mistral achieves better claim recall, we observe generators paired to it achieves better faithfulness, indicating generators are all capable to identify and leverage information in context. Similarly, hallucination and self-knowledge are both reduced as well.

**Retriever Recall Trades-off with Generator Noise Sensitivity.** Claim recall for a retriever characterizes the coverage of all information necessary to produce ground-truth answer. In practice, however, because of the fixed-size chunking strategy, retrieved relevant chunks may inevitably also carry over noise as part of the context. As retriever claim recall increases, all generators become more sensitive to such noise, which can be explained as their faithfulness to certain context is not discriminative enough. This observation shows that generators' capability to precisely leverage relevant context is still a challenge.

**Relevant Noise Sensitivity is More Challenging.** For every baseline RAG system, there's an apparent gap between its relevant and irrelevant noise sensitivity. In correlation to the last paragraph, it further enhance the point that generators demonstrate a chunk-level faithfulness. It means a relevant

chunk is trusted as a whole, while an irrelevant one only has minimal impact. This subtle yet significant distinction supports and explains the importance of the quality and specification of the database for a RAG system.

**Open-Source Models are Worse at Distinguishing Accurate Information from Noise.** GPT-4 has both higher context utilization and lower noise sensitivity than the other three open source models. Open source models are faithful but tend to trust the context blindly especially when retrieval gets better. This observation raises the need for improving open source models' reasoning ability.

### 4.4 Diagnosis on RAG Settings for Improvements

Guided by observations in Sec. 4.3, we modify settings commonly tuned in RAG systems that may lead to improvements, diagnose their working mechanisms with RAGCHECKER metrics, and provide suggestions for improvements on certain aspects. We experiment with different numbers of chunks, chunk sizes, chunk overlap ratios, and generation prompts. We highlight our main findings and suggestions as below, please refer to Appendix F for detailed analysis and results.

**More Context Enhances Faithfulness.** Increasing the number ($k$) and size of chunks improves the recall of more useful information (*claim recall* 61.5→77.6 with $k$ 5→20, 70.3→77.6 with size 150→300). Consequently, this provides more context for the generators to be more faithful to (*faithfulness* 88.1→92.2 with $k$ 5→20, 91.2→92.2 with size 150→300), though at the same time they also become more sensitive to additional noise (*noise sensitivity* 34.0→35.4 with $k$ 5→20, 34.5→35.4 with size 150→300). Improvements in the overall performance (*F1* 51.7→53.4 with $k$ 5→20, 52.6→53.4 with size 150→300) indicates benefits from more context.

**Explicit Requirements in Prompts Affect Generation Preferences.** When prompts introduces explicit requirements for better *faithfulness*, *context utilization*, and lower *noise sensitivity*, generators show improvements in *faithfulness* (92.2→93.6), but struggle with the subtle tension between *context utilization* (59.2→63.7) and *noise sensitivity* (35.4→38.1).

**Chunk Overlap Does Not Matter a Lot.** The chunk overlap ratio is usually set to be non-zero to help generators better utilize surrounding information and identify chunks with coherent logic. However, it minimally affects generation performance, as retrieving more chunks sharing similar useful information (increased *context precision* 69.3→71.1) does not necessarily increase the total amount of retrieved useful information (comparable *claim recall* 77.8→78.1).

**Suggestions to RAG Builders**

Improving the retriever is an effective way to enhance overall performance. While a better embedding model leads to improvements in both *precision* and *recall*, moderately increasing the number and size of chunks improves *recall* and thus *F1* with minimal efforts in practice. Note that the effect saturates as the total amount of relevant information is fixed, so they need not be too large for a balanced cost-performance. On the other hand, given a limited number of context, larger chunk sizes with fewer chunks are preferred for better *context precision*. However, when targeting better *context utilization* or reduced *noise sensitivity*, opposite adjustments should be made to alleviate the influence of noise.

When tuning the generator, the trilemma of *context utilization*, *noise sensitivity*, and *faithfulness* makes it difficult to improve all aspects simultaneously. RAG builders should prioritize certain aspects in the prompt based on their targets, user preferences and the generator's capability.

## 5   Conclusion

This paper presents RAGCHECKER , a novel evaluation framework designed for RAG systems. We validate our comprehensive suite of metrics, both overall and modular, through rigorous human assessments, demonstrating a strong correlation with evaluations conducted by human annotators. We have undertaken a detailed evaluation of eight distinct RAG systems using these metrics, yielding pivotal insights into the behaviors of the retriever and generator components and the trade-offs inherent in RAG system designs. These findings not only deepen our understanding of RAG system architectures but also furnish critical guidance for future advancements in RAG applications.

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

# A    Details for Benchmark Curation

In this section, we introduce the benchmark datasets and the curation process for RAG evaluation. This benchmark datasets are derived from existing open-domain question answering (ODQA) datasets, including RobustQA [9], KIWI [51], ClapNQ [34], and NovelQA [47]. However, most of the ground truth answers in existing ODQA datasets are short answers, while the answers provided by modern LLM-based RAG systems tend to be long-form answers. Therefore, we repurpose the ODQA datasets by eliminating overly simple questions and converting the short answers into long-form answers to match the capabilities of current RAG systems. The statistics of the benchmark are summarized in Tab. 1. In the rest of this section, we describe the datasets we use and the curation process for each domain.

## A.1    Data Sources

**RobustQA**    We choose 7 domains from RobustQA's collection of datasets: Biomedical, Finance, Lifestyle, Recreation, Technology, Science, and Writing. For the Biomedical domain, following RobustQA, we employ the BioASQ [43] dataset, which contains human expert-written question-answer pairs and ground truth documents based on abstracts of articles from PubMed. We use the test sets for Task b from 2014 to 2023 and the corpus of v.2022 to construct the benchmark. We keep QA pairs whose answers are relatively long (more than 50 words), obtaining 511 QA pairs for the biomedical domain. The other 6 domains are sourced from FiQA [26] and LoTTE [36], each of their question is annotated with a list of short answers that are spans of ground truth passages. We convert the short answers to long-form answers using GPT-4 and only keep the generated answers with no hallucinations, as checked by RefChecker. Finally, we sample 500 examples for each domain.

**ClapNQ**    Derived from NaturalQuestions (NQ) [17], an ODQA dataset based on Wikipedia, ClapNQ has long-form answers annotated for a subset of NQ for evaluating RAG. We employ the dev set of ClapNQ in our benchmark and take the annotated long-form answers as the ground truth.

**KIWI**    is constructed by asking LLMs research questions about a set of NLP papers and guiding the LLMs to reach satisfactory long-form answers. The authors validated the quality of the generated answers by rating them as "good", "neutral", or "bad". We take the answers labeled "good" as the ground truth answers and query the full text of the papers from S2ORC [23] as the corpus. As a result, we obtain 71 QA pairs and 429 papers as the corpus.

**NovelQA**    is a benchmark for question answering over long novels containing over 100K tokens on average. Originally designed for benchmarking long-context LLMs, we repurpose it for evaluating RAG. In contrast with the other domains, each question in NovelQA is associated with a single novel, so when we use this dataset for RAG, we constrain the retrieval to within the corresponding novel. We select 19 copyright-free novels and convert the corresponding short answers to long-form answers following the same process for RobustQA.

## A.2    Long-form Answer Generation

We employ GPT-4 (`gpt-4-turbo-2024-04-09`) to convert the human annotated short answers to long-form answers in the dataset of RobustQA and NovelQA. For RobustQA, the short answers are spans of the annotated ground truth passages, we take all the annotated short answers and the corresponding passages in the prompt and ask GPT-4 to convert them to one single long-form answer. For NovelQA, we take the human written evidences as the ground truth passage content and the human written short answers for the long-form answer generation. The prompt is shown in Fig. 2.

For quality control, we ask GPT-4 to generate the passage IDs associated with the long-form answer. We use RefChecker to check whether all the claims of a long-form answer are entailed by these passages, and we only keep the long-form answers that meet this criteria. The RefChecker we used here are described in Appendix G.

```
Here is a question and human annotated short answers, the short
answers are extracted from a passage, you should help me to
convert the short answers to a long-form answer based on the
provided passage. There could be more than one annotation, so you
should identify the best answers and merge them. The long-form
answer should only depend on the provide information, you should
not hallucinate anything.

### Question
{question}
### Annotations
#### Annotation 1
[Passage ID]: {passage_id}
[Passage Content]: {passage_content}
[Short Answers]:
{answers}

#### Annotation 2
[Passage ID]: {passage_id}
[Passage Content]: {passage_content}
[Short Answers]:
{answers}

...

Your should output the converted long-form answer and the passage
IDs you used. Always follow this format for your response:
[Long-form answer]: <the content of the long-form answer>
[Passage IDs]: ID1, ID2, ...
```

Figure 2: The prompt used for converting short answers to long-form answers for the domains of Novel, Finance, Lifestyle, Recreation, Technology, Science, and Writing.

### A.3 Corpus Downsampling for Science and Biomedical Domains

In addition to long-form answer generation, we also perform downsampling for the corpora of Science and Biomedical domains as they are much larger than the others, with over 1 million documents each. Building indexes for a dense retriever is very costly for large corpora, so we downsample these domains to lower the evaluation cost for the community. For the biomedical domain, we first use BM25 retriever to obtain top 400 documents for each question. The subsampled corpus is formed by combining all documents from the retriever with annotated relevant documents from the datasets. Based on our initial study, we observe that the BM25 retriever yeild competitive performance against the dense retriever, so we decide to only use the BM25 retriever for downsampling purpose to save compuation cost. For the science domain, we leverage both the BM25 retriever and e5-mistral-7b-instruct based dense retriever to obtain document candidates. Specifically, we retrieve the top 200 documents from both retrievers (400 documents in total before deduplication). Similarly, the combination of all documents from the retrievers and annotated relevant documents from datasets forms the downsampled corpus.

### A.4 License of The Datasets

The annotations from RobustQA, ClapNQ and NovelQA are under Apache-2.0 License. The corpora of Finance and annotations of KIWI are under CC-BY-SA-4.0. BioASQ is under CC BY 2.5 license. The license for the corpora of LoTTE are not specified.

## B  The complete formula for all metrics

Denote the model response as $m$, the ground truth answer as $gt$, and the retrieved chunks as $\{\text{chunk}_j\}$. Leveraging RefChecker, we decompose the text into a set of claims $\{c_i\}$ and assess whether a specific claim $c_i$ can entail ($\in$) or not entail ($\notin$) a given reference text $Ref$, where $Ref$ may represent $m$, $gt$, or $\{\text{chunk}_j\}$. We assign an entailment label to each ground-truth claim relative to a chunk, and subsequently classify these chunks into relevant chunks $\{\text{r-chunk}_j\}$ and irrelevant chunks

$\{\text{irr-chunk}_j\}$. Specifically, $\text{chunk}_j$ is considered relevant if it contains at least one claim $c_i^{(gt)}$ such that $c_i^{(gt)} \in \text{chunk}_j$.

In accordance with the definitions provided in Section 3.3, we compute each metric using the following formulations:

## B.1 Overall Metrics

$$\text{Precision} = \frac{|\{c_i^{(m)} \mid c_i^{(m)} \in gt\}|}{|\{c_i^{(m)}\}|}$$

$$\text{Recall} = \frac{|\{c_i^{(gt)} \mid c_i^{(gt)} \in m\}|}{|\{c_i^{(gt)}\}|}$$

## B.2 Retriever Metrics

$$\text{Claim Recall} = \frac{|\{c_i^{(gt)} \mid c_i^{(gt)} \in \{\text{chunk}_j\}\}|}{|\{c_i^{(gt)}\}|}$$

$$\text{Context Precision} = \frac{|\{\text{r-chunk}_j\}|}{k}$$

## B.3 Generator Metrics

$$\text{Faithfulness} = \frac{|\{c_i^{(m)} \mid c_i^{(m)} \in \{\text{chunk}_j\}\}|}{|\{c_i^{(m)}\}|}$$

$$\text{Relevant Noise Sensitivity} = \frac{|\{c_i^{(m)} \mid c_i^{(m)} \notin gt \text{ and } c_i^{(m)} \in \{\text{r-chunk}_j\}\}|}{|\{c_i^{(m)}\}|}$$

$$\text{Irrelevant Noise Sensitivity} = \frac{|\{c_i^{(m)} \mid c_i^{(m)} \notin gt \text{ and } c_i^{(m)} \in \{\text{irr-chunk}_j\}\}|}{|\{c_i^{(m)}\}|}$$

$$\text{Hallucination} = \frac{|\{c_i^{(m)} \mid c_i^{(m)} \notin gt \text{ and } c_i^{(m)} \notin \{\text{chunk}_j\}\}|}{|\{c_i^{(m)}\}|}$$

$$\text{Self-knowledge} = \frac{|\{c_i^{(m)} \mid c_i^{(m)} \in gt \text{ and } c_i^{(m)} \notin \{\text{chunk}_j\}\}|}{|\{c_i^{(m)}\}|}$$

$$\text{Context Utilization} = \frac{|\{c_i^{(gt)} \mid c_i^{(gt)} \in \{\text{chunk}_j\} \text{ and } c_i^{(gt)} \in m\}|}{|\{c_i^{(gt)} \mid c_i^{(gt)} \in \{\text{chunk}_j\}\}|}$$

## C Details of Meta Evaluation

In the meta evaluation, we ask 10 annotators compare two responses from the RAG system for each instance in the meta evaluation dataset. Seven of the annotators are in-house annotators, and three of them are graduate students. We pay the students 15 USD per hour and totally cost 255 dollars.

Annotators are required to choose their preference from five options: significantly better, slightly better, tie, slightly worse, or significantly worse. The annotation is based on three metrics: correctness, completeness, and overall assessment. The annotation interface with instructions are shown in Fig. 3

To make sure the human evaluation to be agnostic to specific evaluation metrics, we provide the annotators with a detailed annotation guideline which contains detailed instruction and 5 examples.

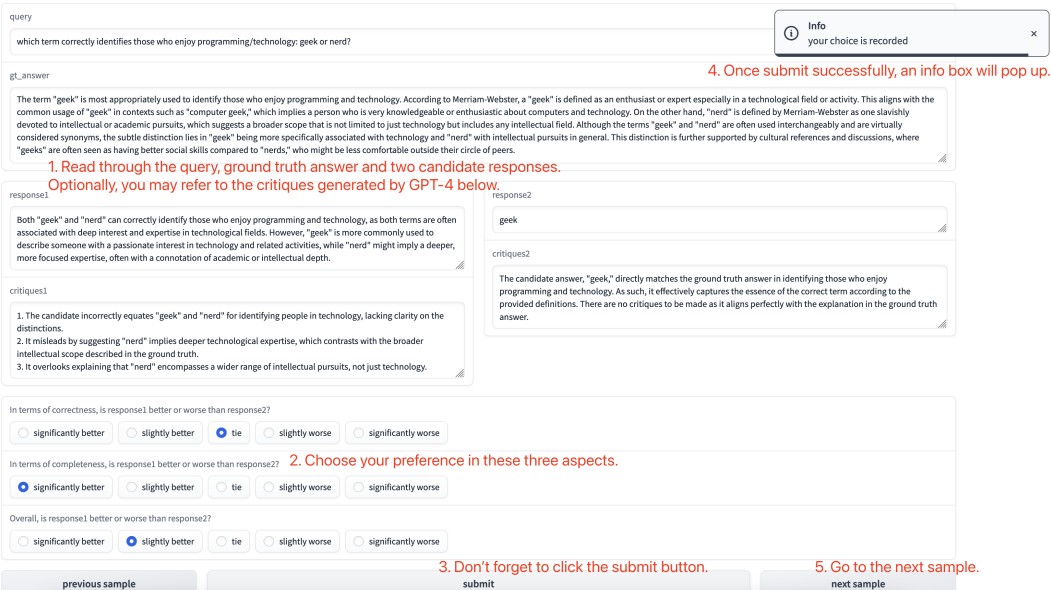

Figure 3: The human annotation interface and instructions of the meta evaluation dataset.

In the UI of the annotation tool, each response is shown with critiques generated by GPT-4 and we ask the annotators to refer to the content of the response and the critiques for labeling. The critiques are generated by prompting GPT-4 to compare the response with ground truth answer to ease the annotation job. In addition, each of the example the guideline are shown with a human-written explanation for the labeling.

The 10 metrics included in the meta evaluation are selected from Trulens [6], RAGAS [5], ARES [35] and CRUD-RAG [25] as explained in Sec. 4.2. Their descriptions are summarized in Tab. 4. As a supplement of Tab. 2, the full correlation results of meta evaluation is shown in Tab. 5. For a detailed comparison between RAGCHECKER and the strongest baseline metric, RAGAS Answer Similarity, we plot the prediction score distribution of two metrics in Fig. 4. From the prediction score distribution and the mean line (dashed line) of the plot, we can observe a stronger correlation of RAGCHECKER than RAGAS Answer Similarity.

## D  Details of the Experiment Setup

**Models in Baseline RAG Systems**  We use the version of `e5-mistral-7b-instruct` for the E5-Mistral retriever. For the generators, we use `gpt-4-turbo-2024-04-09` version for GPT-4, `Llama3-8B-Instruct` for Llama3-8B and `Llama3-70B-Instruct` for Llama3-70B, and `Mixtral-8x7B-Instruct-v0.1` for Mixtral-8x7B.

We adopt OpenSearch[6] as the tool to implement the inverted index for BM25 and the approximate KNN search for dense retrieval. We use a g5.48xlarge instance with 8 NVIDIA A10G GPUs on AWS for inference of open-source models. We split documents in the corpus to chunks of 300 tokens with an overlap ratio of 0.2 by default. We use the tokenizer of E5-Mistral for both retrievers to control the chunking. For each query, top-20 chunks ranked by retrievers are used as context for LLM generation. The default prompt for all generators is shown in Fig. 5. We set the generation temperature to 0.0 (deterministic) and the maximum generation length to 2,048 tokens when calling proprietary LLMs.

## E  Detailed Experiment Results

The detailed evaluation results for all our benchmark datasets can be found in Tab. 6 to Tab. 15.

---

[6]https://opensearch.org/

Table 4: Summary of the metrics included in the meta evaluation.

| Baseline | Metric | Description |
|---|---|---|
| TruLens | Groundedness | Assesses the overlap between each statement in the response and the provided context using an LLM. |
| | Answer Relevance | Prompts an LLM to give a relevance score between the response and question. |
| RAGAS | Faithfulness | Measures the proportion of claims in the response that can be inferred from the context. |
| | Answer Relevance | Computes the mean cosine similarity between the original question and a series of LLM-generated questions derived from the response and context. |
| | Answer Similarity | Measures the semantic similarity between the response and the ground truth answer based on text-embedding-ada-002 [28]. |
| | Answer Correctness | Quantifies both the semantic similarity and the factual overlap between the response and the ground truth answer. |
| ARES | Answer Faithfulness | Prompts an LLM to determine whether the response is faithful to the context. |
| | Answer Relevance | Prompts an LLM to measure whether the response addresses all aspects of the question and provides only correct information from the context. |
| CRUD-RAG | Recall | Computes the ratio of all questions generated from ground truth answers that can be answered by response. |
| | Precision | Evaluates if the generated text is accurate and consistent with the ground truth answer. |

Table 5: Full Correlation results with Human Evaluation of Correctness, Completeness, and Overall Assessment

| Baseline | Metric | Correctness | | Completeness | | Overall Assessment | |
|---|---|---|---|---|---|---|---|
| | | Pearson | Spearman | Pearson | Spearman | Pearson | Spearman |
| BLEU | BLEU-avg | 38.89 | 35.32 | 32.13 | 21.85 | 35.14 | 29.42 |
| ROUGE | ROUGE-L | 31.75 | 31.72 | 47.88 | 45.67 | 43.10 | 43.21 |
| BERTScore | BERTScore | 30.34 | 27.05 | 37.93 | 40.05 | 33.51 | 35.57 |
| TruLens | Groundedness | 21.11 | 18.21 | 14.01 | 6.02 | 19.45 | 14.42 |
| TruLens | Answer Relevance | 35.01 | 27.37 | 37.24 | 37.91 | 35.15 | 33.59 |
| ARES | Answer Relevance | 18.63 | 16.84 | 20.13 | 18.13 | 17.81 | 16.26 |
| ARES | Answer Faithfulness | 9.46 | 7.60 | 10.25 | 8.99 | 8.80 | 7.58 |
| RAGAS | Faithfulness | 8.22 | 7.53 | 4.90 | 1.19 | 7.83 | 5.55 |
| RAGAS | Answer Correctness | 39.11 | 36.30 | 36.42 | 36.04 | 38.01 | 37.14 |
| RAGAS | Answer Similarity | 41.07 | 43.21 | 53.16 | **61.35** | 48.31 | 57.23 |
| RAGAS | Answer Relevance | 11.59 | 8.19 | 9.39 | 13.57 | 10.27 | 11.83 |
| CRUD-RAG | Precision | 20.73 | 15.67 | 25.58 | 20.33 | 25.59 | 19.63 |
| CRUD-RAG | Recall | 30.93 | 27.13 | 45.11 | 43.76 | 41.25 | 39.71 |
| RAGChecker | Same metric as human | **49.66** | **46.95** | **60.67** | 58.11 | **61.93** | **60.90** |
| Human | Annotator sanity check | 63.67 | 59.19 | 71.91 | 68.36 | 70.09 | 68.89 |

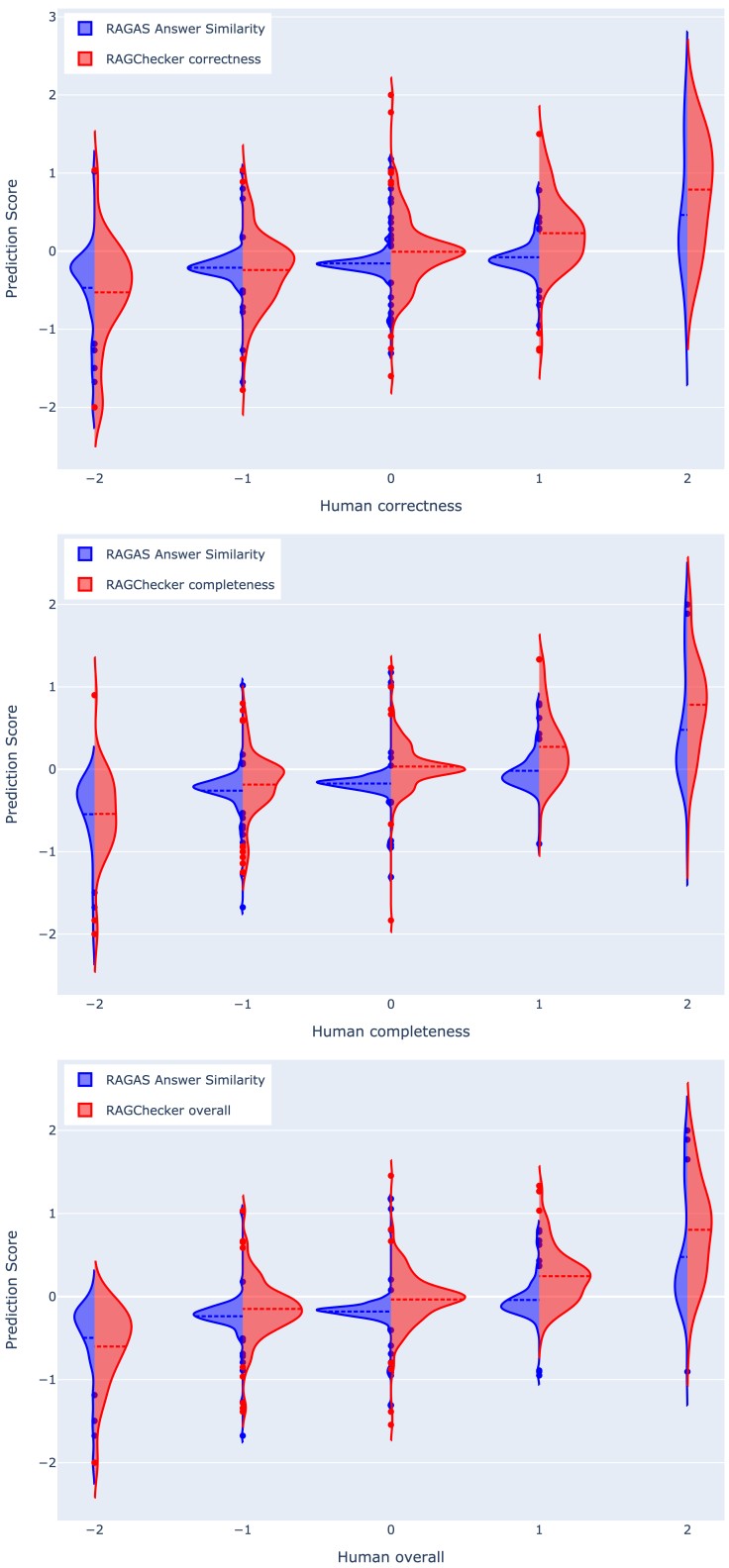

Figure 4: Comparison of prediction score distribution between RAGCHECKER and RAGAS Answer Similarity. Each point in the plot represents an instance in the meta evaluation dataset, where the x-axis is the human preference label under corresponding aspect and y-axis is the prediction score of RAGCHECKER and RAGAS Answer Similarity. The distribution of prediction score is represented by the colored area and the dashed line is the mean line.

```
Please answer the given question based on the context.

<context>
<content>
{chunk_1}
</content>
<content>
{chunk_2}
</content>
...
<content>
{chunk_k}
</content>
</context>

Question: {question}

Please answer the question and tag your answer with <answer></answer>.
```

Figure 5: The default prompt used for response generation in the main experiments for the 8 RAG baseline systems.

## F  Diagnosis on RAG for Improvements

We modify hyper-parameters commonly tuned in RAG systems to observe performance variance under the metrics defined by RAGCHECKER . We focus on how RAGCHECKER explains this variance and provides tuning suggestions for improvements on certain aspects. In this section, we evaluate three RAG baselines (BM25_GPT-4, E5-Mistral_GPT-4, and E5-Mistral_Llama3-70B) across three domains with increasing difficulty: Writing, Finance, and KIWI. We use our default settings (Appendix D) in main experiments as controls. We experiment with different numbers of chunks selected as context $k \in \{5,10,20\}$, different chunk size $\{150,300,600\}$[7], different chunk overlap ratio $\{0.0,0.2,0.4\}$, and different generation prompts.

**More Context Enhances Faithfulness**    Top-k selection and chunk size both balance the amount of noise and useful information presented to the generator, but in different manners. Corresponding results are demonstrated in Fig. 6 and Fig. 7. Increasing $k$ adds more context that could be less relevant, while increasing chunk size provides more surrounding context of relevant facts. Thus *context precision* decreases with larger $k$ but increases with larger chunk sizes. Despite this, they both lead to better *claim recall* in Retrieval.

Generators tend to be more faithful when provided with more context, though this trend is less pronounced for Llama3, which already exhibits high faithfulness. *Context utilization* generally worsens with more context due to increasing noise, leading to higher *relevant noise sensitivity*.

Overall, the end-to-end RAG performance is slightly better with more context, primarily due to improved *recall*. We recommend moderately increasing the two parameters for more faithful generation, noting that saturation occurs at high values as the amount of useful information is limited. Given a limited context length, a larger chunk size with a smaller k is preferred, especially for easier datasets (Finance, Writing). This is evident when comparing a chunk size of 150 with $k$=20 against a chunk size of 300 with $k$=10.

**Explicit Requirements in Prompts Affect Generation Preferences**    To validate the effect of the generation prompt, we added more detailed requirements to guide the generation for better *faithfulness*, *context utilization*, and lower *noise sensitivity*. The optimized prompt is shown in Fig. 9.

As shown in Fig. 8, we observed a general improvement in *context utilization*. However, as a counterpart to *context utilization*, *noise sensitivity* generally worsened. It demonstrates the difficulty of meeting all prompt requirements when there are subtle tension between them.

For the two generators, GPT-4 generally showes improvements in metrics related to faithfulness (*hallucination*, *self-knowledge*, *faithfulness*), whereas Llama3 does not exhibit the same behavior. This aligns with our previous observation (Sec. 4.3) that Llama3 already performs well on *faithfulness*,

---

[7]We omit chunk size of 600 for E5-Mistral_Llama3-70B due to the limited 8K context window of Llama3.

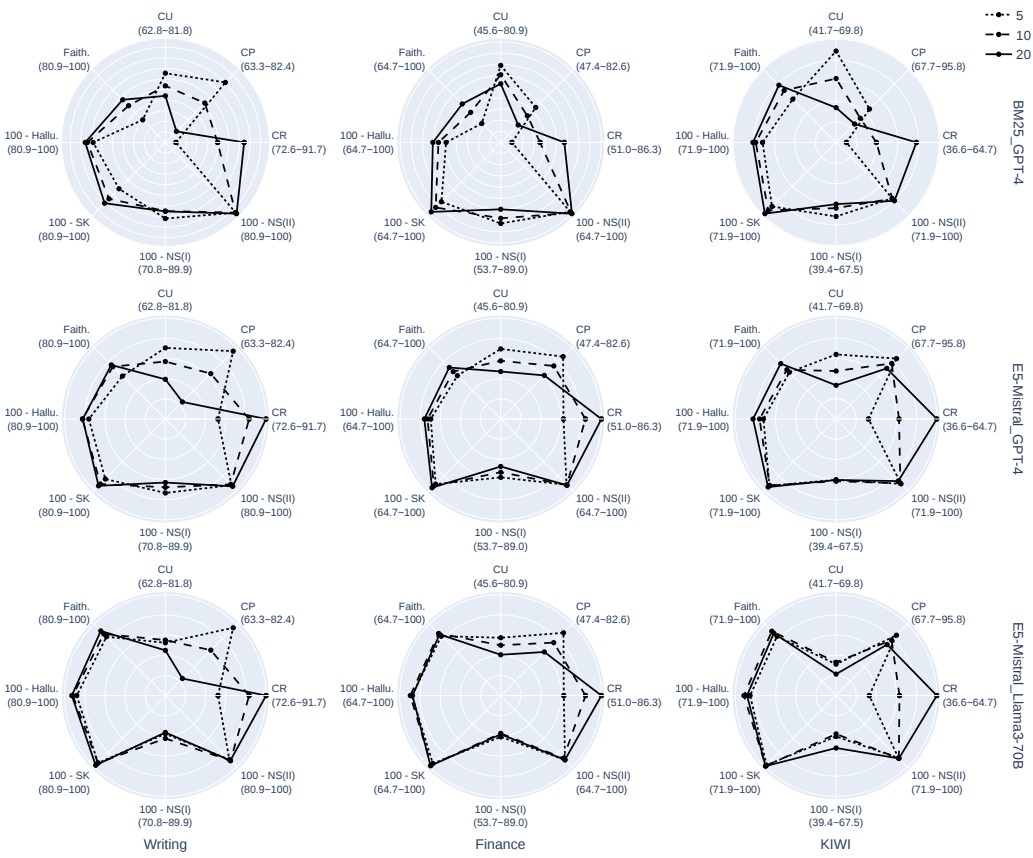

Figure 6: Diagnosis on Top-k Selection

while GPT-4 tends to rely on self-knowledge without explicit requirements. Consequently, there is a steady improvement in overall *F1* for GPT-4 when switched to the optimized prompt, while the difference for Llama3 is negligible.

RAG builders can optimize prompts by combining performance on modular metrics provided by RAGCHECKER with user preferences and generator capabilities on different aspects.

**Chunk Overlap Does Not Matter a Lot**    Chunk overlap ratio between adjacent chunks is usually set to be non-zero to help the generator better utilize surrounding information and identify chunks with coherent logic, thus alleviating the impact of hard splits in significant semantics.

According to our results in Fig. 10, higher overlap ratios generally lead to improved *context precision*. However, this does not necessarily translate to an increase in the total amount of useful information retrieved. This phenomenon can be attributed to the retrieval of more chunks that contain the same segment of useful information. Consequently, we observed that overlap ratio adjustments do not have a significant impact on other performance metrics in a consistent and obvious manner. This suggests that the overlap ratio may not require extensive tuning in practice.

# G  Performance Validation of RefChecker with Llama3 Extractor and Checker

We use Llama3-70B-Instruct for the extractor and checker in RefChecker. To validate the effectiveness of this combination, we test its performance on the RefChecker benchmark. As shown in Tab. 16, Llama 3 based RefChecker outperforms the best purely open-sourced combinations reported in the RefChecker paper in all the three context settings.

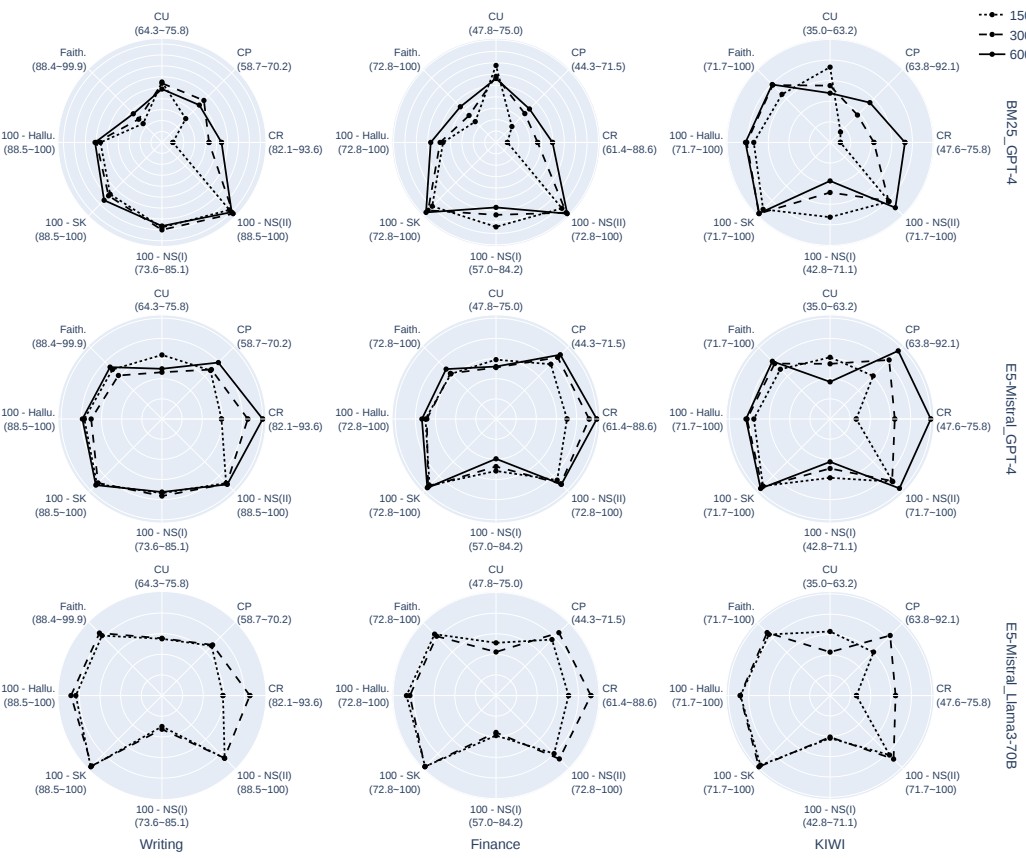

Figure 7: Diagnosis on Chunk Size

# H Limitations

While RAGCHECKER provides a comprehensive evaluation framework for RAG systems, it has a few limitations that should be acknowledged and addressed in future research.

First, the diagnostic metrics for the retriever component are less insightful compared to those for the generator. The retrieval metrics primarily focus on the recall of ground truth claims and precision of retrieved context, but they may not fully capture the nuances and complexities of the retrieval process. Developing more sophisticated metrics that consider factors such as the information density, diversity and coherence of the retrieved context could provide deeper insights into the retriever's performance.

Second, the metrics proposed in RAGCHECKER do not differentiate between Neutral and Contradiction checking results from RefChecker when evaluating the generated responses. These two types of results may have different impacts on the final response quality, and treating them equally could lead to an incomplete assessment. Future work should explore ways to incorporate the distinction between neutral and contradiction results into the evaluation metrics, potentially assigning different weights or penalties based on their severity.

Finally, the evaluation benchmark used in this study is curated based on existing text-only datasets and is limited to English queries and corpus. While this allows for a focused evaluation of RAG systems, it may not fully represent the diverse range of tasks and languages that RAG systems can be applied to. Expanding the benchmark to include datasets from different modalities (e.g., images, audio) and languages would provide a more comprehensive assessment of RAG systems' capabilities and generalization. Additionally, creating benchmark datasets specifically designed for evaluating RAG systems, rather than repurposing existing ones, could help to better capture the unique challenges and requirements of this task.

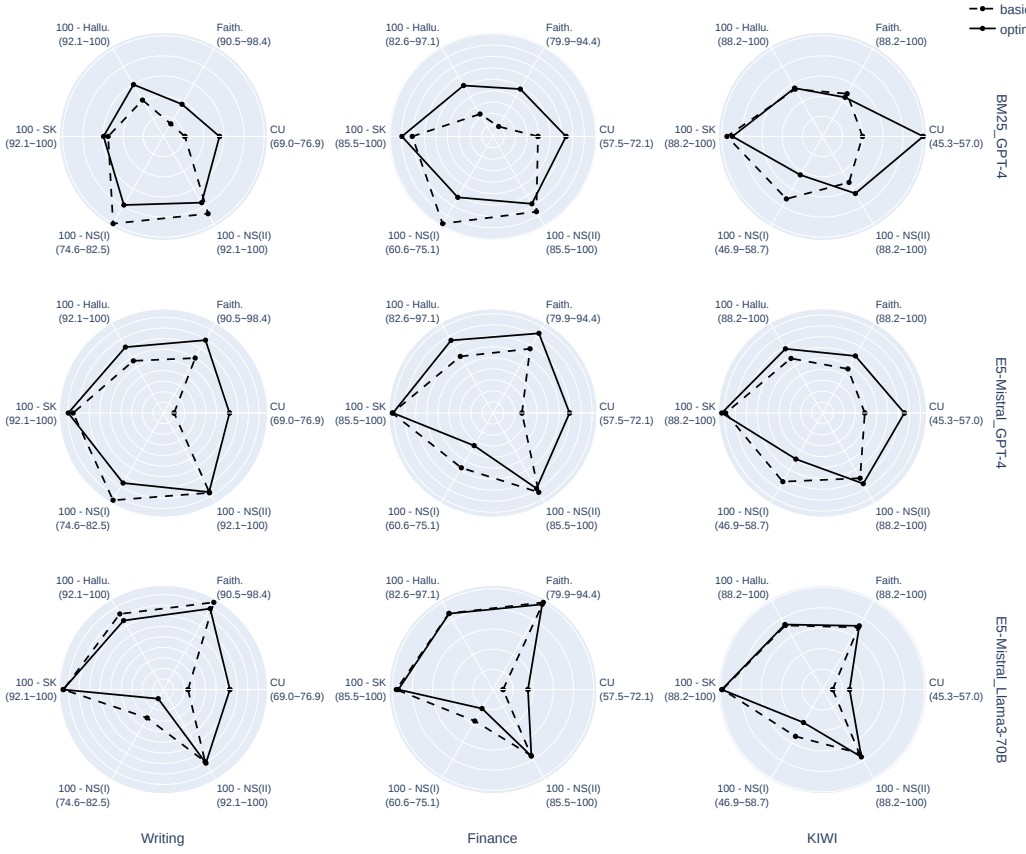

Figure 8: Diagnosis on Generation Prompts

By refining the diagnostic metrics, incorporating the impact of different checking results, and expanding the evaluation benchmark, researchers can gain an even more comprehensive understanding of RAG systems' performance and identify targeted areas for improvement.

# I  Potential Negative Societal Impacts

The RAGCHECKER evaluation framework, while beneficial for assessing RAG systems, could inadvertently lead to several negative societal impacts. There is a risk that developers may focus on optimizing for RAGCHECKER's specific metrics to the detriment of broader utility and ethical considerations. The computational and financial requirements to meet RAGCHECKER standards could disadvantage smaller organizations, potentially centralizing innovation among well-resourced entities. Moreover, an overreliance on quantitative measures might neglect qualitative factors like user experience and ethical implications.

Table 6: Evaluation results for different RAG systems on ClapNQ dataset

| RAG systems | Overall | | | Retriever | | Generator | | | | | | #Claim |
|---|---|---|---|---|---|---|---|---|---|---|---|---|
| | Prec.↑ | Rec.↑ | F1↑ | CR↑ | CP↑ | CU↑ | NS(I)↓ | NS(II)↓ | Hallu.↓ | SK↓ | Faith.↑ | |
| BM25_GPT-4 | 56.9 | 50.0 | 46.7 | 81.1 | 41.3 | 56.4 | 29.4 | 5.9 | 7.5 | 2.2 | 90.3 | 8 |
| BM25_Llama3-8b | 49.6 | 48.6 | 42.2 | 81.1 | 41.3 | 55.2 | 31.9 | 7.5 | 10.8 | 2.0 | 87.2 | 10 |
| BM25_Llama3-70b | 56.9 | 48.7 | 45.3 | 81.1 | 41.3 | 55.7 | 30.1 | 7.1 | 5.9 | 1.6 | 92.4 | 7 |
| BM25_Mixtral-8x7b | 47.9 | 49.6 | 42.1 | 81.1 | 41.3 | 55.8 | 36.9 | 7.3 | 6.9 | 2.3 | 90.9 | 9 |
| E5-Mistral_GPT-4 | 59.7 | 51.1 | 47.9 | 81.5 | 43.6 | 59.9 | 31.1 | 3.8 | 5.4 | 2.3 | 92.3 | 9 |
| E5-Mistral_Llama3-8b | 50.4 | 50.9 | 43.5 | 81.5 | 43.6 | 59.4 | 33.2 | 6.4 | 10.0 | 1.5 | 88.5 | 10 |
| E5-Mistral_Llama3-70b | 58.7 | 52.8 | 48.1 | 81.5 | 43.6 | 61.4 | 32.0 | 5.1 | 4.2 | 2.1 | 93.7 | 8 |
| E5-Mistral_Mixtral-8x7b | 51.1 | 54.4 | 45.7 | 81.5 | 43.6 | 63.2 | 37.0 | 5.2 | 5.5 | 1.5 | 93.0 | 10 |

```
You are an accurate and reliable AI assistant capable of answering
questions using external documents. Always be faithful to the provided
documents and leverage relevant, accurate information from them as
much as possible. Be aware that external documents might contain noisy
or factually incorrect data. Apply critical reasoning to discern and
use the correct information from these sources.

<context>
<content>
{chunk_1}
</content>
<content>
{chunk_2}
</content>
...
<content>
{chunk_k}
</content>
</context>

Question: {question}

Please answer the question and tag your answer with <answer></answer>.
```

Figure 9: The optimized prompt for response generation. In this prompt, we explicitly instruct the LLMs to be faithful to the context and identify relevant information as possible.

Table 7: Evaluation results for different RAG systems on NovelQA dataset

| RAG systems | Overall | | | Retriever | | Generator | | | | | | #Claim |
|---|---|---|---|---|---|---|---|---|---|---|---|---|
| | Prec.↑ | Rec.↑ | F1↑ | CR↑ | CP↑ | CU↑ | NS(I)↓ | NS(II)↓ | Hallu.↓ | SK↓ | Faith.↑ | |
| BM25_GPT-4 | 71.0 | 56.2 | 56.4 | 82.1 | 42.6 | 64.9 | 17.6 | 5.4 | 6.1 | 2.2 | 91.7 | 4 |
| BM25_Llama3-8b | 60.2 | 47.8 | 45.9 | 82.1 | 42.6 | 55.2 | 23.1 | 7.1 | 9.6 | 1.5 | 88.8 | 3 |
| BM25_Llama3-70b | 65.0 | 51.8 | 51.9 | 82.1 | 42.6 | 59.6 | 21.4 | 7.5 | 6.1 | 2.1 | 91.8 | 3 |
| BM25_Mixtral-8x7b | 56.0 | 50.2 | 46.0 | 82.1 | 42.6 | 58.4 | 24.8 | 6.4 | 10.9 | 2.3 | 86.8 | 4 |
| E5-Mistral_GPT-4 | 69.4 | 56.2 | 55.7 | 82.7 | 45.1 | 63.6 | 19.4 | 6.1 | 5.1 | 1.7 | 93.2 | 4 |
| E5-Mistral_Llama3-8b | 58.7 | 48.1 | 45.7 | 82.7 | 45.1 | 55.1 | 23.5 | 8.1 | 9.2 | 1.5 | 89.3 | 3 |
| E5-Mistral_Llama3-70b | 64.5 | 50.2 | 49.6 | 82.7 | 45.1 | 56.9 | 23.7 | 5.5 | 6.0 | 1.5 | 92.4 | 3 |
| E5-Mistral_Mixtral-8x7b | 54.2 | 48.3 | 43.6 | 82.7 | 45.1 | 54.7 | 29.6 | 6.9 | 7.5 | 1.6 | 90.9 | 4 |

Table 8: Evaluation results for different RAG systems on RobustQA - Writing dataset

| RAG systems | Overall | | | Retriever | | Generator | | | | | | #Claim |
|---|---|---|---|---|---|---|---|---|---|---|---|---|
| | Prec.↑ | Rec.↑ | F1↑ | CR↑ | CP↑ | CU↑ | NS(I)↓ | NS(II)↓ | Hallu.↓ | SK↓ | Faith.↑ | |
| BM25_GPT-4 | 76.3 | 63.6 | 66.0 | 86.3 | 64.3 | 70.0 | 17.5 | 1.0 | 5.1 | 4.0 | 90.9 | 10 |
| BM25_Llama3-8b | 65.0 | 59.7 | 57.7 | 86.3 | 64.3 | 66.1 | 26.0 | 1.8 | 6.2 | 2.3 | 91.4 | 10 |
| BM25_Llama3-70b | 72.2 | 62.1 | 63.6 | 86.3 | 64.3 | 68.4 | 23.1 | 1.5 | 3.2 | 2.2 | 94.7 | 8 |
| BM25_Mixtral-8x7b | 67.0 | 60.1 | 59.8 | 86.3 | 64.3 | 66.1 | 25.2 | 1.5 | 4.0 | 2.2 | 93.8 | 8 |
| E5-Mistral_GPT-4 | 77.1 | 65.0 | 67.3 | 91.7 | 66.3 | 69.0 | 17.9 | 1.2 | 3.8 | 1.3 | 94.9 | 10 |
| E5-Mistral_Llama3-8b | 67.4 | 62.8 | 60.6 | 91.7 | 66.3 | 66.8 | 25.5 | 2.2 | 4.5 | 0.6 | 95.0 | 9 |
| E5-Mistral_Llama3-70b | 73.1 | 65.7 | 66.2 | 91.7 | 66.3 | 70.1 | 23.5 | 1.9 | 1.5 | 0.5 | 98.0 | 9 |
| E5-Mistral_Mixtral-8x7b | 66.4 | 62.2 | 61.3 | 91.7 | 66.3 | 66.4 | 26.3 | 2.0 | 3.2 | 0.4 | 96.4 | 9 |

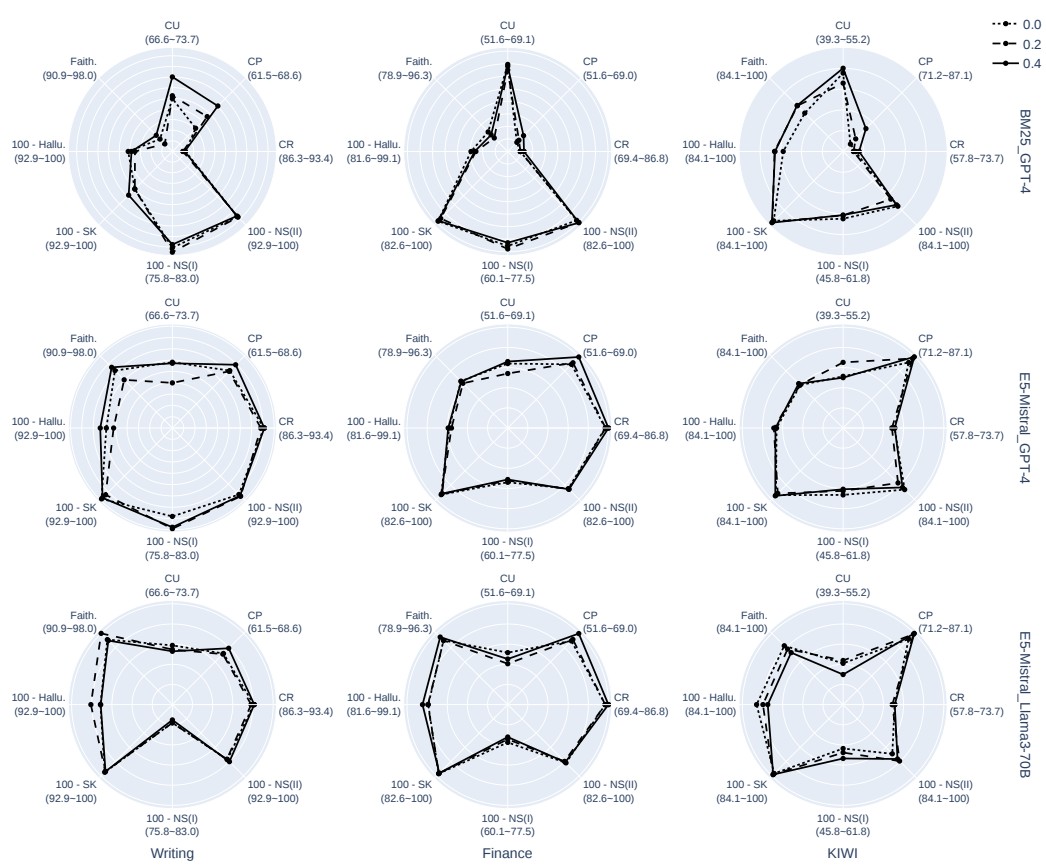

Figure 10: Diagnosis on Chunk Overlap Ratio

Table 9: Evaluation results for different RAG systems on RobustQA - BioASQ dataset

| RAG systems | Overall | | | Retriever | | Generator | | | | | | #Claim |
|---|---|---|---|---|---|---|---|---|---|---|---|---|
| | Prec.↑ | Rec.↑ | F1↑ | CR↑ | CP↑ | CU↑ | NS(I)↓ | NS(II)↓ | Hallu.↓ | SK↓ | Faith.↑ | |
| BM25_GPT-4 | 66.1 | 43.2 | 46.5 | 82.1 | 39.4 | 51.0 | 25.0 | 5.4 | 3.4 | 0.7 | 95.9 | 10 |
| BM25_Llama3-8b | 64.1 | 34.5 | 38.5 | 82.1 | 39.4 | 41.0 | 21.6 | 5.8 | 5.7 | 0.5 | 93.8 | 7 |
| BM25_Llama3-70b | 70.7 | 35.4 | 42.1 | 82.1 | 39.4 | 41.8 | 20.3 | 6.0 | 2.9 | 0.6 | 96.4 | 7 |
| BM25_Mixtral-8x7b | 60.6 | 43.7 | 45.0 | 82.1 | 39.4 | 50.7 | 28.3 | 6.5 | 4.5 | 0.9 | 94.6 | 9 |
| E5-Mistral_GPT-4 | 65.7 | 44.0 | 46.7 | 84.4 | 45.2 | 50.5 | 26.2 | 5.9 | 2.2 | 0.3 | 97.5 | 10 |
| E5-Mistral_Llama3-8b | 65.1 | 36.0 | 40.0 | 84.4 | 45.2 | 41.4 | 22.5 | 5.8 | 3.7 | 0.3 | 96.0 | 8 |
| E5-Mistral_Llama3-70b | 69.9 | 37.3 | 43.5 | 84.4 | 45.2 | 43.0 | 22.2 | 5.7 | 2.2 | 0.4 | 97.3 | 7 |
| E5-Mistral_Mixtral-8x7b | 58.2 | 44.5 | 44.6 | 84.4 | 45.2 | 50.3 | 31.6 | 7.0 | 2.9 | 0.7 | 96.4 | 10 |

Table 10: Evaluation results for different RAG systems on RobustQA - Finance dataset

| RAG systems | Overall | | | Retriever | | Generator | | | | | | #Claim |
|---|---|---|---|---|---|---|---|---|---|---|---|---|
| | Prec.↑ | Rec.↑ | F1↑ | CR↑ | CP↑ | CU↑ | NS(I)↓ | NS(II)↓ | Hallu.↓ | SK↓ | Faith.↑ | |
| BM25_GPT-4 | 54.0 | 50.1 | 48.2 | 69.4 | 52.1 | 62.3 | 26.8 | 3.9 | 15.4 | 4.7 | 79.9 | 15 |
| BM25_Llama3-8b | 44.0 | 42.6 | 39.3 | 69.4 | 52.1 | 55.0 | 35.5 | 6.6 | 13.5 | 2.5 | 84.0 | 14 |
| BM25_Llama3-70b | 53.8 | 43.6 | 44.5 | 69.4 | 52.1 | 56.7 | 34.3 | 5.2 | 6.5 | 2.1 | 91.4 | 10 |
| BM25_Mixtral-8x7b | 46.4 | 41.3 | 39.3 | 69.4 | 52.1 | 53.0 | 37.4 | 6.4 | 6.7 | 2.2 | 91.1 | 11 |
| E5-Mistral_GPT-4 | 56.0 | 54.6 | 51.8 | 86.3 | 67.4 | 60.2 | 31.7 | 2.7 | 9.5 | 1.4 | 89.1 | 15 |
| E5-Mistral_Llama3-8b | 46.9 | 50.1 | 43.9 | 86.3 | 67.4 | 55.5 | 38.4 | 4.6 | 9.5 | 1.1 | 89.4 | 14 |
| E5-Mistral_Llama3-70b | 56.0 | 51.4 | 49.9 | 86.3 | 67.4 | 57.5 | 35.2 | 3.9 | 4.9 | 0.6 | 94.4 | 12 |
| E5-Mistral_Mixtral-8x7b | 49.0 | 49.0 | 45.0 | 86.3 | 67.4 | 54.7 | 39.0 | 5.3 | 3.9 | 0.6 | 95.5 | 12 |

Table 11: Evaluation results for different RAG systems on RobustQA - Lifestyle dataset

| RAG systems | Overall | | | Retriever | | Generator | | | | | | #Claim |
|---|---|---|---|---|---|---|---|---|---|---|---|---|
| | Prec.↑ | Rec.↑ | F1↑ | CR↑ | CP↑ | CU↑ | NS(I)↓ | NS(II)↓ | Hallu.↓ | SK↓ | Faith.↑ | |
| BM25_GPT-4 | 63.3 | 50.5 | 53.0 | 70.2 | 47.0 | 64.8 | 24.0 | 2.7 | 10.0 | 6.0 | 84.0 | 12 |
| BM25_Llama3-8b | 49.7 | 44.8 | 43.8 | 70.2 | 47.0 | 59.2 | 33.2 | 6.1 | 11.1 | 2.2 | 86.8 | 12 |
| BM25_Llama3-70b | 59.6 | 44.4 | 47.5 | 70.2 | 47.0 | 58.5 | 30.7 | 3.8 | 5.9 | 2.4 | 91.8 | 9 |
| BM25_Mixtral-8x7b | 52.8 | 43.5 | 44.1 | 70.2 | 47.0 | 56.8 | 34.5 | 4.9 | 6.8 | 2.3 | 90.9 | 10 |
| E5-Mistral_GPT-4 | 66.4 | 57.6 | 58.9 | 89.7 | 64.0 | 62.5 | 26.2 | 2.2 | 5.2 | 1.4 | 93.4 | 13 |
| E5-Mistral_Llama3-8b | 54.6 | 55.0 | 51.3 | 89.7 | 64.0 | 59.7 | 34.6 | 4.4 | 6.3 | 0.6 | 93.0 | 14 |
| E5-Mistral_Llama3-70b | 64.8 | 56.7 | 57.3 | 89.7 | 64.0 | 61.7 | 29.3 | 3.3 | 2.6 | 0.7 | 96.7 | 12 |
| E5-Mistral_Mixtral-8x7b | 56.2 | 51.8 | 50.5 | 89.7 | 64.0 | 56.5 | 34.8 | 4.6 | 3.0 | 0.8 | 96.2 | 11 |

Table 12: Evaluation results for different RAG systems on RobustQA - Recreation dataset

| RAG systems | Overall | | | Retriever | | Generator | | | | | | #Claim |
|---|---|---|---|---|---|---|---|---|---|---|---|---|
| | Prec.↑ | Rec.↑ | F1↑ | CR↑ | CP↑ | CU↑ | NS(I)↓ | NS(II)↓ | Hallu.↓ | SK↓ | Faith.↑ | |
| BM25_GPT-4 | 62.9 | 51.9 | 53.1 | 70.4 | 37.5 | 65.7 | 21.4 | 4.6 | 11.1 | 5.2 | 83.7 | 11 |
| BM25_Llama3-8b | 50.6 | 45.1 | 43.1 | 70.4 | 37.5 | 58.1 | 31.0 | 10.1 | 8.1 | 1.8 | 90.2 | 11 |
| BM25_Llama3-70b | 58.1 | 45.7 | 46.5 | 70.4 | 37.5 | 60.2 | 30.4 | 6.3 | 4.9 | 2.0 | 93.1 | 8 |
| BM25_Mixtral-8x7b | 50.5 | 44.3 | 42.4 | 70.4 | 37.5 | 56.2 | 33.1 | 8.3 | 6.7 | 1.8 | 91.5 | 9 |
| E5-Mistral_GPT-4 | 62.4 | 57.0 | 56.1 | 85.1 | 51.1 | 64.2 | 27.8 | 4.1 | 5.7 | 1.5 | 92.8 | 12 |
| E5-Mistral_Llama3-8b | 51.1 | 52.5 | 47.3 | 85.1 | 51.1 | 59.4 | 34.1 | 9.2 | 5.3 | 0.7 | 94.0 | 13 |
| E5-Mistral_Llama3-70b | 60.4 | 53.7 | 52.7 | 85.1 | 51.1 | 60.3 | 30.8 | 6.2 | 2.6 | 0.7 | 96.7 | 10 |
| E5-Mistral_Mixtral-8x7b | 52.1 | 51.8 | 47.9 | 85.1 | 51.1 | 58.6 | 34.1 | 8.2 | 3.9 | 0.6 | 95.5 | 11 |

Table 13: Evaluation results for different RAG systems on RobustQA - Science dataset

| RAG systems | Overall | | | Retriever | | Generator | | | | | | #Claim |
|---|---|---|---|---|---|---|---|---|---|---|---|---|
| | Prec.↑ | Rec.↑ | F1↑ | CR↑ | CP↑ | CU↑ | NS(I)↓ | NS(II)↓ | Hallu.↓ | SK↓ | Faith.↑ | |
| BM25_GPT-4 | 58.7 | 52.6 | 51.8 | 71.3 | 62.6 | 66.6 | 27.6 | 2.4 | 11.3 | 4.3 | 84.5 | 14 |
| BM25_Llama3-8b | 47.9 | 45.2 | 41.7 | 71.3 | 62.6 | 58.2 | 32.2 | 5.3 | 14.2 | 1.7 | 84.1 | 14 |
| BM25_Llama3-70b | 55.8 | 45.0 | 45.5 | 71.3 | 62.6 | 57.7 | 33.0 | 4.5 | 6.7 | 1.6 | 91.7 | 10 |
| BM25_Mixtral-8x7b | 51.5 | 45.1 | 44.0 | 71.3 | 62.6 | 58.5 | 34.8 | 5.4 | 7.1 | 1.8 | 91.1 | 10 |
| E5-Mistral_GPT-4 | 57.9 | 55.0 | 53.0 | 85.0 | 71.8 | 61.5 | 31.5 | 2.3 | 8.4 | 1.5 | 90.2 | 15 |
| E5-Mistral_Llama3-8b | 48.8 | 48.5 | 43.5 | 85.0 | 71.8 | 54.8 | 39.1 | 4.6 | 6.6 | 0.6 | 92.8 | 13 |
| E5-Mistral_Llama3-70b | 56.7 | 51.3 | 49.6 | 85.0 | 71.8 | 57.7 | 36.1 | 3.9 | 3.2 | 0.5 | 96.3 | 11 |
| E5-Mistral_Mixtral-8x7b | 54.5 | 49.2 | 47.4 | 85.0 | 71.8 | 55.3 | 37.1 | 3.7 | 4.4 | 0.6 | 95.0 | 11 |

Table 14: Evaluation results for different RAG systems on RobustQA - Technology dataset

| RAG systems | Overall | | | Retriever | | Generator | | | | | | #Claim |
|---|---|---|---|---|---|---|---|---|---|---|---|---|
| | Prec.↑ | Rec.↑ | F1↑ | CR↑ | CP↑ | CU↑ | NS(I)↓ | NS(II)↓ | Hallu.↓ | SK↓ | Faith.↑ | |
| BM25_GPT-4 | 57.5 | 48.6 | 48.9 | 69.5 | 63.8 | 63.4 | 28.1 | 3.1 | 11.2 | 4.3 | 84.5 | 14 |
| BM25_Llama3-8b | 47.2 | 46.3 | 42.1 | 69.5 | 63.8 | 61.2 | 36.3 | 5.8 | 10.1 | 1.7 | 88.2 | 14 |
| BM25_Llama3-70b | 55.9 | 44.4 | 45.5 | 69.5 | 63.8 | 59.2 | 35.6 | 4.4 | 4.0 | 1.6 | 94.4 | 10 |
| BM25_Mixtral-8x7b | 51.7 | 43.4 | 42.9 | 69.5 | 63.8 | 58.2 | 36.5 | 5.4 | 5.6 | 1.7 | 92.7 | 11 |
| E5-Mistral_GPT-4 | 59.9 | 55.0 | 53.6 | 83.7 | 76.4 | 63.3 | 31.8 | 2.3 | 6.0 | 1.2 | 92.8 | 15 |
| E5-Mistral_Llama3-8b | 47.9 | 51.1 | 44.8 | 83.7 | 76.4 | 59.0 | 40.4 | 5.6 | 5.9 | 0.5 | 93.6 | 15 |
| E5-Mistral_Llama3-70b | 56.9 | 53.5 | 50.7 | 83.7 | 76.4 | 62.1 | 36.7 | 3.8 | 2.6 | 0.3 | 97.1 | 13 |
| E5-Mistral_Mixtral-8x7b | 52.6 | 51.5 | 47.9 | 83.7 | 76.4 | 59.6 | 40.5 | 3.4 | 3.2 | 0.2 | 96.6 | 12 |

Table 15: Evaluation results for different RAG systems on KIWI dataset

| RAG systems | Overall | | | Retriever | | Generator | | | | | | #Claim |
|---|---|---|---|---|---|---|---|---|---|---|---|---|
| | Prec.↑ | Rec.↑ | F1↑ | CR↑ | CP↑ | CU↑ | NS(I)↓ | NS(II)↓ | Hallu.↓ | SK↓ | Faith.↑ | |
| BM25_GPT-4 | 42.8 | 30.0 | 32.4 | 57.8 | 72.5 | 49.1 | 45.0 | 6.2 | 6.0 | 0.7 | 93.3 | 18 |
| BM25_Llama3-8b | 43.0 | 24.7 | 26.6 | 57.8 | 72.5 | 39.6 | 41.8 | 4.8 | 9.1 | 1.7 | 89.2 | 16 |
| BM25_Llama3-70b | 42.7 | 27.4 | 31.0 | 57.8 | 72.5 | 43.8 | 45.2 | 6.8 | 5.2 | 0.7 | 94.0 | 18 |
| BM25_Mixtral-8x7b | 40.2 | 21.6 | 23.8 | 57.8 | 72.5 | 35.5 | 51.0 | 5.8 | 3.1 | 0.3 | 96.6 | 13 |
| E5-Mistral_GPT-4 | 45.5 | 34.0 | 36.0 | 64.6 | 86.7 | 49.0 | 44.9 | 4.0 | 5.5 | 1.5 | 93.0 | 20 |
| E5-Mistral_Llama3-8b | 47.2 | 27.8 | 29.8 | 64.6 | 86.7 | 39.1 | 43.4 | 4.2 | 4.6 | 0.3 | 95.2 | 15 |
| E5-Mistral_Llama3-70b | 45.2 | 30.9 | 34.0 | 64.6 | 86.7 | 45.3 | 47.5 | 3.7 | 3.6 | 0.3 | 96.1 | 18 |
| E5-Mistral_Mixtral-8x7b | 36.7 | 23.1 | 23.5 | 64.6 | 86.7 | 32.2 | 55.0 | 4.9 | 2.9 | 0.5 | 96.7 | 14 |

Table 16: Performance of RefChecker on the RefChecker benchmark using Llama 3 70B Instruct as both the extractor and checker. We compare the results with the best performed purely open-sourced combinations reported in the RefChecker paper.

| | Accuracy | Fact. F1 | Non-Fact. F1 | Pearson | Spearman |
|---|---|---|---|---|---|
| Zero Context | | | | | |
| Mistral-SFT + RepC | 89.38 | 80.43 | 92.72 | 77.14 | 76.74 |
| Llama3 + Llama3 | **91.89** | **83.06** | **94.67** | **81.77** | **80.83** |
| Noisy Context | | | | | |
| Mistral-SFT + NLI | 70.82 | 75.12 | **64.72** | 52.21 | 45.61 |
| Llama3 + Llama3 | **71.75** | **76.69** | 64.15 | **57.67** | **50.31** |
| Accurate Context | | | | | |
| Mistral-SFT + AlignScore | 74.12 | 81.6 | 56.38 | 46.34 | 43.22 |
| Llama3 + Llama3 | **78.35** | **84.87** | **61.92** | **59.48** | **52.03** |

## Supplementary Material Checklist

1. Submission introducing new datasets must include the following in the supplementary materials:

   (a) Dataset documentation and intended uses. Recommended documentation frameworks include datasheets for datasets, dataset nutrition labels, data statements for NLP, and accountability frameworks. [Yes] In the Datasheets file

   (b) URL to website/platform where the dataset/benchmark can be viewed and downloaded by the reviewers. [Yes] The data can be downloaded from a list of AWS s3 links. We put the download scripts in the zip file and show the command in README.md

   (c) URL to Croissant metadata record documenting the dataset/benchmark available for viewing and downloading by the reviewers. You can create your Croissant metadata using e.g. the Python library available here: https://github.com/mlcommons/croissant [No]

   (d) Author statement that they bear all responsibility in case of violation of rights, etc., and confirmation of the data license. [Yes] We bear all responsibility in case of violation of rights, etc., and confirmation of the data license.

   (e) Hosting, licensing, and maintenance plan. The choice of hosting platform is yours, as long as you ensure access to the data (possibly through a curated interface) and will provide the necessary maintenance. [Yes] We will continually provide hosting, licensing, and maintenance.

2. To ensure accessibility, the supplementary materials for datasets must include the following:

   (a) Links to access the dataset and its metadata. This can be hidden upon submission if the dataset is not yet publicly available but must be added in the camera-ready version. In select cases, e.g when the data can only be released at a later date, this can be added afterward. Simulation environments should link to (open source) code repositories. [Yes] The data can be downloaded from a list of AWS s3 links. We put the download scripts in the zip file and show the command in README.md

   (b) The dataset itself should ideally use an open and widely used data format. Provide a detailed explanation on how the dataset can be read. For simulation environments, use existing frameworks or explain how they can be used. [Yes] The data are contained in JSON files

   (c) Long-term preservation: It must be clear that the dataset will be available for a long time, either by uploading to a data repository or by explaining how the authors themselves will ensure this. [Yes]

   (d) Explicit license: Authors must choose a license, ideally a CC license for datasets, or an open source license for code (e.g. RL environments). [Yes] CC for data and Apache 2.0 for code.

   (e) Add structured metadata to a dataset's meta-data page using Web standards (like schema.org and DCAT): This allows it to be discovered and organized by anyone. If you use an existing data repository, this is often done automatically. [Yes]

   (f) Highly recommended: a persistent dereferenceable identifier (e.g. a DOI minted by a data repository or a prefix on identifiers.org) for datasets, or a code repository (e.g. GitHub, GitLab,...) for code. If this is not possible or useful, please explain why. [Yes] We will release it to GitHub after a reviewing process.

3. For benchmarks, the supplementary materials must ensure that all results are easily reproducible. Where possible, use a reproducibility framework such as the ML reproducibility checklist, or otherwise guarantee that all results can be easily reproduced, i.e. all necessary datasets, code, and evaluation procedures must be accessible and documented. [Yes]

4. a brief discussion on the main concerns raised by previous reviewers and how you addressed them: [N/A]

