# OpenReview forum: "RAGChecker: A Fine-grained Framework for Diagnosing Retrieval-Augmented Generation"
_NeurIPS.cc/2024/Datasets_and_Benchmarks_Track — NeurIPS 2024 Track Datasets and Benchmarks Poster_

### Official Review · Reviewer_Kzxa · 2024-06-25
**RAGChecker: A Fine-grained Framework for Diagnosing Retrieval-Augmented Generation**

**Rating:** 8
**Confidence:** 4
**Correctness:** Yes
**Clarity:** Yes

**Review:**

Please refer to Strengths and Limitations.

**Strengths:**

This paper presents RAGCHECKER, a novel framework offering fine-grained evaluation for RAG systems with new diagnostic metrics to identify error sources.

Meta evaluation shows RAGCHECKER significantly outperforms other metrics in correlating with human judgments.

Evaluating 8 RAG systems across 10 domains reveals insights such as the trade-off between retrieval improvement and noise, and the tendency of open-source models to overly trust context.

**Additional Feedback:**

no

**Documentation:**

yes

**Limitations:**

How is the overall performance in Table 3 evaluated? Accuracy is also an important metric for evaluating a RAG system, and it should be provided as well

**Opportunities For Improvement:**

This paper's dataset is sourced from existing datasets. As a dataset-focused work, I would have preferred to see the contribution of a new dataset. However, this paper provides many valuable perspectives on RAG analysis, which is also highly valuable.

**Relation To Prior Work:**

yes

**Summary And Contributions:**

Evaluating Retrieval-Augmented Generation (RAG) systems is challenging due to their modular nature and measurement reliability. This paper introduces RAGCHECKER, a fine-grained evaluation framework with metrics for both retrieval and generation. RAGCHECKER shows better correlation with human judgments than other metrics. It was used to evaluate 8 RAG systems, revealing key patterns and trade-offs in design, and guiding the development of more effective RAG systems.

---

> ### Author Rebuttal · Authors · 2024-08-17
>
> Dear Reviewer,
>
> We sincerely appreciate your thoughtful review and valuable feedback on our paper. We are grateful for your positive assessment. We address your comments as below:
>
> **Dataset Contribution**
>
> Thanks for your comments on potential improvements. While our RAG benchmark repurposes existing datasets, we've conducted additional work such as generating long-form answers for RobustQA, ClapNQ, and NovelQA so that they are appropriate for the specific RAG scenario. Our primary focus, however, is on developing a fine-grained diagnostic evaluation framework, particularly the novel metrics we've introduced.
>
> **Overall Performance Evaluation in Table 3**
>
> The overall performance (precision, recall, F1) follows the process described in Section 3.3.1. Sorry for the confusion, we acknowledge that this could be made more explicit and will clarify this in our revision.
>
> **Inclusion of accuracy as a metric**
>
> We didn't include accuracy in the main results as it may be overly rigorous for long LLM-generated responses, counting only 100% semantically matched responses as correct. There could be multiple plausible expressions of the golden answer. For your reference, we've included the accuracy results for the evaluated RAG systems below:
>
> | RAG System             | F1    | Accuracy |
> |------------------------|-------|----------|
> | BM25_GPT-4             | 54.7  | 19.4     |
> | BM25_Llama3-8b         | 47.4  | 18.0     |
> | BM25_Llama3-70b        | 50.8  | 22.0     |
> | BM25_Mixtral-8x7b      | 48.4  | 17.8     |
> | E5-Mistral_GPT-4       | 57.1  | 19.4     |
> | E5-Mistral_Llama3-8b   | 50.6  | 19.0     |
> | E5-Mistral_Llama3-70b  | 54.8  | 22.1     |
> | E5-Mistral_Mixtral-8x7b| 51.2  | 16.0     |
>
> These results illustrate why we hesitated to include accuracy as a primary metric, given the relatively low scores due to the strict matching criteria. However, we greatly value your perspective on this matter. Do you have any suggestions for adapting the accuracy metric to better suit long-form response generation in RAG systems?
>
> We appreciate your insights and are committed to improving our paper based on your feedback. Thank you again for your valuable review and the time you've invested in helping us enhance our work.

---

### Official Review · Reviewer_EEuk · 2024-07-25
**Reviews**

**Rating:** 6
**Confidence:** 3

**Review:**

Pros:
1. Guided by the idea that different users pay attention to different aspects of RAG systems, RagChecker extends the evaluation of RAG systems to retriever and generator modules.
2. RagChecker introduces more metrics in claim-level, which makes the evaluation and diagnosis more fine-grained and interpretable.
3. The authors perform experiments with human annotated datasets to verify that RagChecker matches human preference significantly better than existing metrics.

Cons:
1. The claim-level evaluation is based on the extraction of claims and the judgment of whether claims are entailed in chunks. This paper does not specify how to extract claims from generated text and how to judge whether claims are entailed in retrieved chunks.
2. The meta-evaluation is the correlation between evaluation model outputs and human annotations. There are annotations from two human annotators. How to calculate the correlation, by averaging the human scores or calculating the correlations separately and then averaging the correlations?
3. Human annotation for meta-evaluation includes correctness, completeness and overall assessment. However, the baseline metrics with descriptions listed in Table 4 have different meanings. Is the correlation analysis meaningful regarding their varying evaluation aspects? And which metrics of RagChecker are used as correctness, completeness and overall assessment to be compared with human annotations?

**Strengths:**

1. With emergent and capable LLMs, RAG is important to supplement LLMs with external knowledge and defend against hallucination. It’s of practical significance to develop a scientific and reasonable RAG evaluation and diagnosis framework.
2. The proposed claim-level metrics are simple yet effective in evaluating the capabilities and defects of RAG system modules. Utilizing RagChecker helps when selecting among RAG systems according to specific capability requirements.

For more strengths, see Pros in ‘Review’.

**Additional Feedback:**

All comments, suggestions, and questions are in ‘Review’ and ‘Opportunities for Improvement’.

**Clarity:**

This paper is well-written and accurately expressed. The proposed evaluation metrics are easy to follow.

**Correctness:**

As an evaluating and diagnosing framework, the evaluation methods and experiment design are appropriate and performed correctly. However, the datasets included in the experiment are somewhat limited. The claims made by diagnosing for improvements are intuitive though common consensus.

**Documentation:**

For the curated datasets, this paper provides sufficient detail on data collection and organization. For the proposed benchmark, this paper includes sufficient detail to support reproducibility.

**Ethics:**

No ethical concerns.

**Limitations:**

The authors analyze the limitations of RagChecker and propose potential solutions to address them, including refining the diagnostic metrics, incorporating the impact of different checking results and expanding the evaluation benchmark.

**Opportunities For Improvement:**

1. To correct some typos. For example, ‘of between’ in Line 93, ‘a apparent gap’ in Line 294.
2. The analyses in Section 4.4 are somewhat straightforward, many of which are common consensus to develop better RAG systems.
3. ‘The trilemma of context utilization, noise sensitivity and faithfulness’ is interesting, which could be discussed further.

For more suggestions, see Cons in ‘Review’.

**Relation To Prior Work:**

Prior work mainly evaluates the essential capabilities of generators only or the end-to-end performance of RAG systems. RagChecker extends the evaluation to claim-level and includes retriever, generator, and overall assessment.

**Summary And Contributions:**

This paper focuses on the comprehensive evaluation and diagnosis for the whole process and each single module of retrieval-augmented generation systems. The authors propose a fine-grained evaluation framework called RagChecker, by which the evaluation of RAG is extended to retriever/generator modules and refined to the claim level. Meta-evaluation is performed to verify that RagChecker has better correlations with human judgments than existing metrics. Furthermore, 8 RAG systems are evaluated using RagChecker to reveal insightful patterns and trade-offs in the design choices of RAG architectures.

---

> ### Author Rebuttal · Authors · 2024-08-17
>
> Dear Reviewer,
>
> We sincerely appreciate your thorough review and insightful comments. We are grateful for the opportunity to address your concerns and clarify aspects of our work.
>
> **Claim Extraction and Entailment Judgment**
>
> We apologize for any lack of clarity. As mentioned in Section 4.1, we adopted the open-sourced framework RefChecker to perform claim extraction and entailment checking. Specifically, it prompts the backbone LLM (Llama3-70B in our experiments) to extract knowledge triplets from the response as claims, and performs the NLI task with each retrieved chunk as the premise and the extracted claims as hypothesis for entailment checking. We will add more details in our later revision.
>
> **Meta-evaluation Correlation Calculation**
>
> We compute the correlation on the concatenated annotations from both human annotators. Specifically, we treat the annotations from different annotators as separate instances in our analysis.
>
> **Correlation Analysis and Metric Comparison**
>
> We acknowledge the varying aspects of the baseline metrics. However, we have tried our best to construct the baselines by selecting the best-performing metrics in these baseline systems (Full results are listed in Table 5, Appendix C). In RagChecker, precision, recall, and F1 correspond to correctness, completeness, and overall assessment, respectively. We will clarify this in the revised paper.
>
> **Typos and Writing Organization**
>
> We appreciate your attention to details. We will conduct a thorough proofreading to address all typos. Regarding the organization, we will restructure our analysis to emphasize novel findings and provide more in-depth discussions on uncommon observations.
>
> We are committed to improving our paper based on your valuable feedback. Thank you for your constructive comments that will help enhance the quality and clarity of our work.

---

> > ### Comment · Reviewer_EEuk · 2024-08-27
> > **Thank you for the responses**
> >
> > Thank you for the responses.

---

### Official Review · Reviewer_dzR1 · 2024-08-01
**A method for evaluation RAG systems**

**Rating:** 7
**Confidence:** 4
**Correctness:** yes
**Clarity:** yes

**Review:**

In this paper, the authors introduce a fine-grained evaluation for RAG. More specifically, they extract claims from the ground truth answer and the response, and design several metrics for evaluating the overall quality, the retriever quality, and the generator quality.

Pros:
- The proposed evaluation framework is more comprehensive than existing evaluation methods, although similar ideas have already been investigated in existing works on evaluating RAG systems.
- The design of the generator metrics is nice.
- The paper is well-written and easy to read.

Cons:
- Leveraging fine-grained level information for RAG evaluation has been studied in existing papers, as indicated by the authors.
- As revealed by existing works, a retriever in a RAG system should not just return "relevant" passages, and it should be optimized for returning information that can help improve generation quality. This means that the correct claim that the original LLM can generate should not necessarily returned by the retriever. By this means, evaluating the retriever solely based on the context precision and claim recall might be problematic.
- The claim extractor and checker are the key factors impacting the evaluation quality. More experiments with different extractors and checkers are expected.

**Strengths:**

- The proposed evaluation framework is more comprehensive than existing evaluation methods, although similar ideas have already been investigated in existing works on evaluating RAG systems.
- The design of the generator metrics is nice.
- The paper is well-written and easy to read.

**Additional Feedback:**

Please check the Review part.

**Documentation:**

yes

**Limitations:**

- Leveraging fine-grained level information for RAG evaluation has been studied in existing papers, as indicated by the authors.
- As revealed by existing works, a retriever in a RAG system should not just return "relevant" passages, and it should be optimized for returning information that can help improve generation quality. This means that the correct claim that the original LLM can generate should not necessarily returned by the retriever. By this means, evaluating the retriever solely based on the context precision and claim recall might be problematic.
- The claim extractor and checker are the key factors impacting the evaluation quality. More experiments with different extractors and checkers are expected.

**Opportunities For Improvement:**

- Leveraging fine-grained level information for RAG evaluation has been studied in existing papers, as indicated by the authors.
- As revealed by existing works, a retriever in a RAG system should not just return "relevant" passages, and it should be optimized for returning information that can help improve generation quality. This means that the correct claim that the original LLM can generate should not necessarily returned by the retriever. By this means, evaluating the retriever solely based on the context precision and claim recall might be problematic.
- The claim extractor and checker are the key factors impacting the evaluation quality. More experiments with different extractors and checkers are expected.

**Relation To Prior Work:**

yes

**Summary And Contributions:**

In this paper, the authors introduce a fine-grained evaluation for RAG. More specifically, they extract claims from the ground truth answer and the response, and design several metrics for evaluating the overall quality, the retriever quality, and the generator quality.

Pros:
- The proposed evaluation framework is more comprehensive than existing evaluation methods, although similar ideas have already been investigated in existing works on evaluating RAG systems.
- The design of the generator metrics is nice.
- The paper is well-written and easy to read.

Cons:
- Leveraging fine-grained level information for RAG evaluation has been studied in existing papers, as indicated by the authors.
- As revealed by existing works, a retriever in a RAG system should not just return "relevant" passages, and it should be optimized for returning information that can help improve generation quality. This means that the correct claim that the original LLM can generate should not necessarily returned by the retriever. By this means, evaluating the retriever solely based on the context precision and claim recall might be problematic.
- The claim extractor and checker are the key factors impacting the evaluation quality. More experiments with different extractors and checkers are expected.

---

> ### Author Rebuttal · Authors · 2024-08-17
>
> Dear Reviewer,
>
> We appreciate your thoughtful feedback and the opportunity to address your concerns.
>
> **Regarding fine-grained information in RAG evaluation**
>
> We acknowledge that fine-grained information has been utilized in some prior work (e.g., RAGAS). However, our contribution lies in proposing a comprehensive evaluation framework that uniquely groups lower-level, fine-grained checking results to assess multiple aspects of RAG systems. This approach provides valuable diagnostic information for system improvement.
>
> **On retriever evaluation based on context precision and claim recall**
>
> Thanks for the insightful comment. We recognize that in real-world RAG systems, generators may require fewer (to reduce context burden through deduplication) or more (for complex reasoning scenarios) than just the "relevant" passages. We acknowledge that the retriever cannot be evaluated based solely on context precision and claim recall in these cases. However, addressing this may require a completely new design for retrieval evaluation, which could be inconsistent with mainstream retrieval systems today. We consider this as future work and currently focus on fine-grained diagnostic evaluation. Additionally, such cases could be evaluated indirectly by comparing generator metrics across different retrievers. We will expand on this concept in later revision.
>
> **Regarding experiments with different claim extractors and checkers**
>
> We appreciate this constructive suggestion. We are collecting results on more extractors and checkers, especially for smaller ones. Results will be shared during the discussion period. We look forward to sharing these results during the discussion period and thank you for your patience.
>
> We believe these clarifications address your concerns while highlighting the value of our work. We welcome any further questions or comments.

---

> > ### Author Rebuttal · Authors · 2024-08-23
> >
> > Dear Reviewer dzR1,
> >
> > Regarding your comment on experiments with different claim extractors and checkers, and also commented by reviewer xC1v, we conducted two experiments for different extractors and checkers and compare their efficency: 1) applying smaller LLMs as the evaluators; 2) joint checking for the claims.
> >
> > **Applying Smaller LLMs as Evaluators**
> >
> > For a fair comparison, we experimented with SOTA LLMs with different sizes:
> > - [Llama-3.1-70B](https://huggingface.co/meta-llama/Meta-Llama-3.1-70B-Instruct) vs  [Llama-3.1-8B](https://huggingface.co/meta-llama/Meta-Llama-3.1-8B)
> > - [Gemma-2 2B](https://huggingface.co/google/gemma-2-2b-it) vs [Gemma-2 9B](https://huggingface.co/google/gemma-2-9b-it). There are bugs for deploying the 27B version of Gemma-2 with [vllm](https://github.com/vllm-project/vllm), so we just compare the 2B and 9B versions here.
> >
> > The results on the meta-evaluation data are as follows. We found that, for a specific series of LLMs, larger models have higher correlations with humans. However, we also found that Gemma-2 9B is comparable with the 70B models of Llama-3 and Llama-3.1.
> >
> > |      | Correctness (Pearson) | Correctness (Spearman)  | Completeness (Pearson) | Completeness (Spearman) | Overall (Pearson) | Overall (Spearman) |
> > | --- | --- | --- | --- |  --- | --- | --- |
> > |RAGAS (text-embedding-ada-002)	|41.07	|43.21	|53.16	|61.35	|48.31	|57.23	|
> > |RAGChecker (Llama-3-70B)	|49.66	|46.95	|60.67	|58.11	|61.93	|60.9	|
> > |RAGChecker (Llama-3.1-70B) |48.32	|46.01	|66.7	|64.06	|64.47	|61.51	|
> > |RAGChecker (Llama-3.1-8B) |34.31	|28.25	|45.96	|41.18	|48.81	|44.38	|
> > |RAGChecker (Gemma-2 2B)	|30.63	|27.54	|34.41	|32.32	|41.85	|34.7	|
> > |RAGChecker (Gemma-2 9B)	|51.59	|46.94	|61.29	|57.48	|61.43	|59.72	|
> >
> >
> > **Joint Checking**
> >
> > We found that the main bottleneck of the efficiency lies in the claim checking process as the claims are checked one-by-one, i.e. one prompt for one claim. To accelerate this process, we experimented with "Joint Checking" of the claims to check more than one claim in one prompt. Joint checking is especially useful when calling APIs of the LLM services due the rate limit of the APIs (e.g. limited request per minute).
> >
> > The following table shows the results of joint checking of 5 claims in one prompt. The results reveal that joint checking with RAGChecker using 70B models still outperforms the strongest baseline RAGAS.
> >
> > |      | Correctness (Pearson) | Correctness (Spearman)  | Completeness (Pearson) | Completeness (Spearman) | Overall (Pearson) | Overall (Spearman) |
> > | --- | --- | --- | --- |  --- | --- | --- |
> > |RAGAS (text-embedding-ada-002) |41.07	|43.21	|53.16	|61.35	|48.31	|57.23	|
> > |RAGChecker (Llama-3-70B)	|49.66	|46.95	|60.67	|58.11	|61.93	|60.9	|
> > |RAGChecker (Llama-3.1-70B) |48.32	|46.01	|66.7	|64.06	|64.47	|61.51	|
> > |Joint Checking (Llama-3-70B)	|45.58	|43.98	|58.4	|55.55	|59.16	|57.12	|
> > |Joint Checking (Llama-3.1-70B)	|48.67	|45	|58.81	|55.03	|60.76	|58.14	|
> > |Joint Checking (Gemma-2 9B)	|37.21	|29.88	|51.99	|48.31	|51.83	|46.35	|
> >
> > **Latency Test**
> >
> > We deploy the LLMs with vllm on a 8xA100 40G instance and test the latency on checking 50 ClapNQ examples. The batch sizes are 50 for claim extraction and 64 for checking. The instance for LLM deployment and the code for checking are on different devices, so the values of the latency include the communication overhead.
> >
> > The results are as follows. Llama-3.1-70B obtains comparable efficiency as Gemma-2 9B when using joint checking.
> >
> > | | Seconds per example	|
> > | --- | --- |
> > |Gemma-2 2B	|2.78 |
> > |Gemma-2 9B	|7.32 |
> > |Gemma-2 9B (joint checking) | 2.58 |
> > |Llama-3.1-70B|23.6 |
> > |Llama-3.1-70B (joint checking) |7.84 |
> >
> > However, when using API calls for the LLM-based extractors and checkers, we may encounter the rate limit issue. To test the efficiency in this scenario, we compare one-by-one checking and joint checking with Llama-3-70B by calling the API on Amazon Bedrock. The rate limit is 400 requests per minute. As the following table shows, joint checking is ~6x faster than one-by-one checking when there is a rate limit for LLM API calling.
> >
> > | | Seconds per example	|
> > | --- | --- |
> > |Llama-3-70B|69.74 |
> > |Llama-3-70B (joint checking) |11.72 |
> >
> >
> > **Suggestions on the Performance and Efficiency Trade-off**
> >
> > We conclude these experiments with the two suggestions:
> > If we deploy the LLM-based extractor/checker ourselves, we can use a smaller but strong LLM (e.g. Gemma-2 9B) with one-by-one checking.
> > If we use API calls, we can use a strong LLM with joint checking.
> >
> > We will add these experimental results and the prompts of joint checking to the paper.
> >
> > Finally, we want to note that the prompt for checking is zero-shot, so there is still room to improve the performance, including prompt engineering and fine-tuning a small LLM for extraction and checking, etc. We leave them as future works.
> >
> >
> > Many thanks

---

### Official Review · Reviewer_xC1v · 2024-08-03
**Novel direction, but the experimental setup is very limited.**

**Rating:** 6
**Confidence:** 4
**Correctness:** The claims are supported by their exp…
**Clarity:** The paper is well written and is easy…

**Review:**

Strength:
1. Such system is on great demand that developers require such fine-grained feedback for RAG system diagnose.
2. The idea of decomposing the answer and documents into claims is interesting and reasonable for the purpose of fine-grained evaluation.
3. The evaluation results reveal insightful suggestions for future RAG designs.

Weakness:
The paper only uses a single model, Llama3-70B, for claim decomposition and evaluation. However, running evaluation with such large model is very expensive. It remains unknown if the method can be applied with a smaller model or cheaper approximation.
Besides, the baselines in the paper are very weak, which uses a single metric for all dimensions. I could understand this work might be the first to propose fine-grained dimension, but for fair comparison, the authors may also want to show the efficiency-performance trade-off when comparing with those basic metrics.

**Strengths:**

1. The idea of decomposing the generated answers and documents into claims is novel for RAG evaluation, which enables the system to conduct more fine-grained evaluation and obtain consistent scores.
2. The fine-grained metrics for generator are very interesting that decouple different failure reasons, which will provide more direct feedback on the generator.
3. The analysis is inspiring for future works to improve the RAG performance.

**Additional Feedback:**

1. Could the authors compare the results using different LLMs and show the efficiency-performance trade-off?
2. Could the author use different models for claim decomposer and evaluator?

**Documentation:**

The documentations are sufficient.

**Ethics:**

No ethical concern.

**Limitations:**

As stated in the weakness in the review section, my main concern is the efficiency of the proposed method and the very limited setup of the evaluator model in the experiments.

**Opportunities For Improvement:**

1. Could the author provide efficiency analysis of the RAGChecker system? Since the author used Llama3-70B as both claim decomposer and model checker, the evaluation process might be very expensive, which will prevent convenient adoption in development.
2. I wonder if using a smaller model, such as 7B or even 1B model could still achieve reasonable performance over other cheap metrics.

**Relation To Prior Work:**

The contribution is novel in a way that it conducts more fine-grained evaluation than prior work.

**Summary And Contributions:**

This paper presents an evaluation benchmark, RAGChecker, for RAG systems for both end-to-end performance and intermediate diagnose.

For end-to-end evaluation, the benchmark computes precision, recall, and F1 scores. For intermediate diagnose, it computes fine-grained scores based on claims extracted by an LLM.

Through comprehensive evaluation, the paper shows RAGChecker presents higher correlation with human evaluation. The analysis indicate the current challenges and potential improvement directions for RAG.

---

> ### Author Rebuttal · Authors · 2024-08-17
>
> Dear Reviewer,
>
> Thank you for your valuable feedback. We appreciate your insights and would like to address your main concerns:
>
> **Use of only Llama3-70B**
>
> We acknowledge the limitations of using a large model. We are currently conducting experiments with smaller models for both extraction and checking. Results will be shared during the discussion period. Thanks in advance for your patience.
>
> **Baseline Strength**
>
> We apologize for any confusion. Our baselines are selected as the best-performing metrics from existing systems (full results are listed in Table 5, Appendix C). The single metric shown in Table 4 for each baseline consistently outperformed others in their systems across all dimensions in our experiments. This finding underscores the need for multi-dimensional evaluation, as proposed in our framework.
>
> We will clarify these points in our revision. Thank you once again for your time and expertise in reviewing our work.

---

> > ### Author Rebuttal · Authors · 2024-08-23
> >
> > Dear Reviewer xC1v,
> >
> > Regarding your comment on efficiency of RAGChecker, we conducted two experiments for improvement: 1) applying smaller LLMs as the evaluators; 2) joint checking for the claims.
> >
> > First of all, we want to highlight that the efficiency issue does not weaken the main contributions of this work. As we will show later, we can mitigate the efficiency problem with  smaller but stronger LLMs as evaluators and better prompting for checking.
> >
> >
> > **Applying Smaller LLMs as Evaluators**
> >
> > For a fair comparison, we experimented with SOTA LLMs with different sizes:
> > - [Llama-3.1-70B](https://huggingface.co/meta-llama/Meta-Llama-3.1-70B-Instruct) vs  [Llama-3.1-8B](https://huggingface.co/meta-llama/Meta-Llama-3.1-8B)
> > - [Gemma-2 2B](https://huggingface.co/google/gemma-2-2b-it) vs [Gemma-2 9B](https://huggingface.co/google/gemma-2-9b-it). There are bugs for deploying the 27B version of Gemma-2 with [vllm](https://github.com/vllm-project/vllm), so we just compare the 2B and 9B versions here.
> >
> > The results on the meta-evaluation data are as follows. We found that, for a specific series of LLMs, larger models have higher correlations with humans. However, we also found that Gemma-2 9B is comparable with the 70B models of Llama-3 and Llama-3.1.
> >
> > |      | Correctness (Pearson) | Correctness (Spearman)  | Completeness (Pearson) | Completeness (Spearman) | Overall (Pearson) | Overall (Spearman) |
> > | --- | --- | --- | --- |  --- | --- | --- |
> > |RAGAS (text-embedding-ada-002)	|41.07	|43.21	|53.16	|61.35	|48.31	|57.23	|
> > |RAGChecker (Llama-3-70B)	|49.66	|46.95	|60.67	|58.11	|61.93	|60.9	|
> > |RAGChecker (Llama-3.1-70B) |48.32	|46.01	|66.7	|64.06	|64.47	|61.51	|
> > |RAGChecker (Llama-3.1-8B) |34.31	|28.25	|45.96	|41.18	|48.81	|44.38	|
> > |RAGChecker (Gemma-2 2B)	|30.63	|27.54	|34.41	|32.32	|41.85	|34.7	|
> > |RAGChecker (Gemma-2 9B)	|51.59	|46.94	|61.29	|57.48	|61.43	|59.72	|
> >
> >
> > **Joint Checking**
> >
> > We found that the main bottleneck of the efficiency lies in the claim checking process as the claims are checked one-by-one, i.e. one prompt for one claim. To accelerate this process, we experimented with "Joint Checking" of the claims to check more than one claim in one prompt. Joint checking is especially useful when calling APIs of the LLM services due the rate limit of the APIs (e.g. limited request per minute).
> >
> > The following table shows the results of joint checking of 5 claims in one prompt. The results reveal that joint checking with RAGChecker using 70B models still outperforms the strongest baseline RAGAS.
> >
> > |      | Correctness (Pearson) | Correctness (Spearman)  | Completeness (Pearson) | Completeness (Spearman) | Overall (Pearson) | Overall (Spearman) |
> > | --- | --- | --- | --- |  --- | --- | --- |
> > |RAGAS (text-embedding-ada-002) |41.07	|43.21	|53.16	|61.35	|48.31	|57.23	|
> > |RAGChecker (Llama-3-70B)	|49.66	|46.95	|60.67	|58.11	|61.93	|60.9	|
> > |RAGChecker (Llama-3.1-70B) |48.32	|46.01	|66.7	|64.06	|64.47	|61.51	|
> > |Joint Checking (Llama-3-70B)	|45.58	|43.98	|58.4	|55.55	|59.16	|57.12	|
> > |Joint Checking (Llama-3.1-70B)	|48.67	|45	|58.81	|55.03	|60.76	|58.14	|
> > |Joint Checking (Gemma-2 9B)	|37.21	|29.88	|51.99	|48.31	|51.83	|46.35	|
> >
> > **Latency Test**
> >
> > We deploy the LLMs with vllm on a 8xA100 40G instance and test the latency on checking 50 ClapNQ examples. The batch sizes are 50 for claim extraction and 64 for checking. The instance for LLM deployment and the code for checking are on different devices, so the values of the latency include the communication overhead.
> >
> > The results are as follows. Llama-3.1-70B obtains comparable efficiency as Gemma-2 9B when using joint checking.
> >
> > | | Seconds per example	|
> > | --- | --- |
> > |Gemma-2 2B	|2.78 |
> > |Gemma-2 9B	|7.32 |
> > |Gemma-2 9B (joint checking) | 2.58 |
> > |Llama-3.1-70B|23.6 |
> > |Llama-3.1-70B (joint checking) |7.84 |
> >
> > However, when using API calls for the LLM-based extractors and checkers, we may encounter the rate limit issue. To test the efficiency in this scenario, we compare one-by-one checking and joint checking with Llama-3-70B by calling the API on Amazon Bedrock. The rate limit is 400 requests per minute. As the following table shows, joint checking is ~6x faster than one-by-one checking when there is a rate limit for LLM API calling.
> >
> > | | Seconds per example	|
> > | --- | --- |
> > |Llama-3-70B|69.74 |
> > |Llama-3-70B (joint checking) |11.72 |
> >
> >
> > **Suggestions on the Performance and Efficiency Trade-off**
> >
> > We conclude these experiments with the two suggestions:
> > If we deploy the LLM-based extractor/checker ourselves, we can use a smaller but strong LLM (e.g. Gemma-2 9B) with one-by-one checking.
> > If we use API calls, we can use a strong LLM with joint checking.
> >
> > We will add these experimental results and the prompts of joint checking to the paper.
> >
> > Finally, we want to note that the prompt for checking is zero-shot, so there is still room to improve the performance, including prompt engineering and fine-tuning a small LLM for extraction and checking, etc. We leave them as future works.
> >
> >
> > Many thanks

---

> > > ### Comment · Reviewer_xC1v · 2024-08-23
> > >
> > > Thank the authors for the detailed explanation. I appreciate it. I have some follow-up questions:
> > > 1. The latency looks high even for the 2B checker. What would be the real-world application cases for this checking process?
> > > 2. I wonder how this could help improve the end-to-end performance of the RAG pipeline, i.e., how people could use it to improve the retrieved documents and the final generation output.

---

> > ### Author Rebuttal · Authors · 2024-08-26
> >
> > Dear Reviewer xC1v,
> >
> > Thank you for your follow-up questions. We've provided detailed responses below:
> >
> > **Q1: The latency looks high even for the 2B checker.**
> >
> > We've expanded our latency test results by breaking down the total latency into individual checking components:
> >
> > - Extraction: Claim extraction for both the response and ground truth
> > - answer2response: Checking claims in the response using the ground truth answer as a reference
> > - response2answer: Checking claims in the ground truth answer using the response as a reference
> > - retrieved2response: Checking claims in the response using retrieved chunks as a reference
> > - retrieved2answer: Checking claims in the ground truth answer using retrieved chunks as a reference
> >
> > The following table illustrates that the main bottleneck lies in "retrieved2response + retrieved2answer". In our experiments, we used **20 retrieved chunks** (each with 300 tokens) per example. Thus, the latency also depends on the number of retrieved chunks in real-world applications.
> >
> > | Model | Extraction | answer2response + response2answer | retrieved2response + retrieved2answer | Total |
> > |-------|------------|-----------------------------------|---------------------------------------|-------|
> > | Gemma-2 2B | 0.1 | 1.2 | 1.48 | 2.78 |
> > | Gemma-2 9B | 0.22 | 0.26 | 6.84 | 7.32 |
> > | Gemma-2 9B (joint checking) | 0.18 | 0.08 | 2.32 | 2.58 |
> > | Llama-3.1-70B | 0.56 | 0.68 | 22.36 | 23.6 |
> > | Llama-3.1-70B (joint checking) | 0.48 | 0.26 | 7.1 | 7.84 |
> >
> > **Q2: What would be the real-world application cases for this checking process?**
> >
> > RAGChecker is designed for **offline evaluation** during the development of RAG systems. We expect developers to gain actionable insights from the diagnostic results to improve their RAG systems. In real-world use cases, these insights are more critical than checking efficiency.
> >
> > **Q3: How could this help improve the end-to-end performance of the RAG pipeline, i.e., how could people use it to improve the retrieved documents and the final generation output?**
> >
> > As described in Section 4.4, RAGChecker can help improve both the retriever and generator modules in RAG. It computes metrics for **diagnosing the system** rather than mitigating errors in a specific output online. In practice, developers can compare metrics using different retrievers and generators to:
> >
> > 1. Determine which retriever/generator works best in their RAG system
> > 2. Identify improvement areas for the retriever, e.g., chunk size, number of retrieved chunks
> > 3. Pinpoint improvement areas for the generator, e.g., revising the system prompt to be faithful to the context, implementing reasoning steps to identify relevant/noisy information in the context
> >
> > We appreciate your time in discussing this work with us. Your questions have been insightful and will help us improve our results and clarify our writing.
> >
> > Thank you.

---

> > ### Author Rebuttal · Authors · 2024-08-29
> >
> > Dear Reviewer xC1v,
> >
> > We conducted additional experiments to compare the latency of RAGChecker with RAGAS. We focused on two metrics that have similar definitions in both frameworks, with and without contexts:
> >
> > 1. Faithfulness
> > 2. Answer Correctness (RAGAS) vs. F1 Score (RAGChecker)
> >
> > Key differences in the evaluation process for these metrics are:
> >
> > - RAGAS includes all contexts and claims in a single prompt, while RAGChecker evaluates contexts individually for a more fine-grained evaluation.
> > - Answer Correctness in RAGAS relies on an embedding model from OpenAI (text-embedding-ada-002). For a fair comparison, we used `gpt-4o-mini` as the evaluator and `text-embedding-ada-002` as the embedding model for both RAGAS and RAGChecker.
> >
> > For the faithfulness metric, we used Llama 3.1 70B Instruct as the LLM for extraction and checking, as Gemma models lack sufficient context length for RAGAS. The results below are based on the 50 ClapNQ examples previously mentioned, tested under identical conditions. Our findings indicate comparable latency between the two frameworks for these metrics.
> >
> > | Framework | Answer Correctness / F1 | Faithfulness |
> > |-----------|-------------------------|--------------|
> > | RAGAS     | 0.8                     | 3.52         |
> > | RAGChecker| 0.24                    | 4.0          |
> >
> > It's important to note that in RAGAS, each metric uses a specific prompt for claim extraction and checking. Consequently, different metrics do not share checking results, which could, in principle, be shared.
> >
> > In contrast, RAGChecker bases all its metrics on four fundamental checking results:
> >
> > 1. answer2response
> > 2. response2answer
> > 3. retrieved2response
> > 4. retrieved2answer
> >
> > Once these four checking results are obtained, we can compute all the metrics defined in RAGChecker.

---

> > > ### Comment · Reviewer_xC1v · 2024-08-30
> > >
> > > Thank the authors for the response, which has adequately addressed my questions. I have raised my score accordingly. The paper could be stronger if the designed system could help identify the key limitations of current SOTA RAG systems, fix the issue based on the diagnose, and achieve new SOTA results in the end-to-end tasks.

---

### Decision · Program_Chairs · 2024-09-26

**Decision:**

Accept (Poster)

**Comment:**

The paper proposes a fine-grained evaluation method for RAG. It evaluates different aspects of RAG system. New metrics at claim-level are introduced. The paper provides new insights to RAG. Experiments with human annotated datasets show that RagChecker matches better human preference than existing metrics.
The paper may help generating more ideas for RAG evaluations.